# Tackling Feature and Sample Heterogeneity in Decentralized Multi-Task Learning: A Sheaf-Theoretic Approach

## Abstract

Federated multi-task learning (FMTL) aims to simultaneously learn multiple related tasks across clients without sharing sensitive raw data. However, in the decentralized setting, existing FMTL frameworks are limited in their ability to capture complex task relationships and handle feature and sample heterogeneity across clients. To address these challenges, we introduce a novel sheaf-theoretic-based approach for FMTL. By representing client relationships using cellular sheaves, our framework can flexibly model interactions between heterogeneous client models. We formulate the sheaf-based FMTL optimization problem using sheaf Laplacian regularization and propose the `Sheaf-FMTL` algorithm to solve it. We show that the proposed framework provides a unified view encompassing many existing federated learning (FL) and FMTL approaches. Furthermore, we prove that our proposed algorithm, `Sheaf-FMTL`, achieves a sublinear convergence rate in line with state-of-the-art decentralized FMTL algorithms. Extensive experiments demonstrate that `Sheaf-FMTL` exhibits communication savings by sending significantly fewer bits compared to decentralized FMTL baselines.

## 1 Introduction

The growing demand for privacy-preserving distributed learning algorithms has steered the research community towards federated learning (FL) (McMahan et al., 2017), a learning paradigm that allows several clients, such as mobile devices or organizations, to cooperatively train a model without revealing their raw data. By aggregating locally computed updates rather than raw data, FL aims to learn a global model that benefits from the different data distributions inherently present across the participating clients. Despite its promise, conventional FL faces significant hurdles when dealing with client data heterogeneity. In fact, while the global model may perform well on average, the statistically heterogeneous clients' data have been shown to affect the model's existence and convergence (Sattler et al., 2020; Li et al., 2020b). These challenges are exacerbated in a decentralized environment where coordination is limited and direct control over the client models is not feasible. Recently, there have been several attempts to bring personalization into FL to learn distinct local models (Wang et al., 2019; Fallah et al., 2020; Hanzely & Richtárik, 2020) since learning a personalized model per client is more suitable than a single global model to tackle data heterogeneity. These models are specifically learned to fit the heterogeneous local data distribution via techniques such as federated multi-task learning (FMTL) (Smith et al., 2017; Dinh et al., 2022) that model the interactions between the different personalized local models.

FMTL generalizes the FL framework by allowing the simultaneous learning of multiple related but distinct tasks across several clients. Unlike traditional FL, which focuses on training a single global model, FMTL acknowledges that different clients may be interested in solving distinct tasks that are related but not identical. By leveraging task relatedness, FMTL aims to improve the performance of individual task models through shared knowledge while maintaining task-specific uniqueness. This approach not only enhances the generalization performance of the models on individual tasks by leveraging shared information but also contributes to tackling the non-independent and identically distributed (non-IID) nature of data across clients. FMTL considers the different objectives and data distributions across clients, customizing models to perform optimally on each task while still benefiting from the federated structure of the problem as well as the similarity between these tasks.

However, existing FMTL frameworks are not without limitations. A major concern is the oversimplified view of task interdependencies, where relationships between tasks are often modeled using simple fixed scalar weights. This approach captures only the basic notion of task relatedness but fails to represent more intricate and higher-order dependencies that may exist among tasks. As a result, these models may overlook subtle interconnections and dynamic patterns of interdependence, leading to suboptimal knowledge sharing and reduced performance in heterogeneous and decentralized environments. For a comprehensive understanding of scenarios where task similarities are naturally defined in vector spaces, kindly refer to Appendix F. Furthermore, a critical issue with the current FMTL frameworks is the assumption that the models have the same size, which limits the applicability of FMTL in the case of different model sizes. Last but not least, to the best of our knowledge, apart from MOCHA (Smith et al., 2017), which requires the presence of a server, the interactions between the tasks are assumed to be known and not learned during training. Therefore, to address these challenges, we seek to answer the following question:

**"How can we effectively model and learn the complex interactions between various tasks/models in an FMTL decentralized setting, even in the presence of different model sizes?"**

To answer this question, the concept of sheaves provides a novel lens through which the interactions between clients in a decentralized FMTL setting can be modeled. The mathematical notion of a sheaf initially invented and developed in algebraic topology, is a framework that systematically organizes local observations in a way that allows one to make conclusions about the global consistency of such observations (Robinson, 2014; 2013; Riess & Ghrist, 2022). As such requirements are a central part of FL problems, it is natural to ask how one could utilize a sheaf-based framework to find an effective solution to the above question. Given an underlying topology of the client relationships, we employ the notion of a cellular sheaf that captures the underlying geometry, enabling a richer and more nuanced multi-task learning environment. Sheaves enable the representation of local models as *sections* over the underlying space, offering a structured way to capture the relationships between tasks/models in FMTL settings. As an inherent feature of the sheaf-based framework, our approach can support heterogeneity over local models. More specifically, our framework naturally facilitates learning models with different model dimensions. Moreover, sheaves are inherently distributed in nature and hence facilitate decentralized training. A sheaf data structure consists of vector spaces and linear mappings between them. In this work, we model the underlying space as a graph, and vector spaces are defined over vertices and edges, capturing pairwise interactions. Crucially, we are required to learn these maps that constitute the sheaf structure, as a decision variable of our problem. Learning these maps is instrumental in comparing heterogeneous models by projecting them onto a common space.

**Contributions.** This paper introduces a novel unified approach that fundamentally rethinks FMTL and gives it a new interpretation by incorporating principles from sheaf theory. In what follows, we summarize our main contributions

- Our proposed framework demonstrates a high degree of flexibility as it addresses the challenges arising from both feature and sample heterogeneity in the context of FMTL exploiting sheaf theory. It may be regarded as a comprehensive and unified framework for FMTL, as it encompasses a multitude of existing frameworks, including personalized FL (Hanzely & Richtárik, 2020), conventional FMTL (Dinh et al., 2022), hybrid FL (Zhang et al., 2024), and conventional FL (McMahan et al., 2017).

- To the best of our knowledge, this is the first work that proposes to solve the FMTL in a decentralized setting while modelling higher-order relationships among heterogeneous clients. Furthermore, unlike existing decentralized FMTL frameworks, we learn the interactions between the clients as part of our optimization framework.

- Our algorithm, coined `Sheaf-FMTL`, exhibits high communication efficiency, as the size of shared vectors among clients is significantly smaller in practice compared to the original models. Furthermore, exchanging a modified version of the clients' models provides an additional layer of privacy.

- A detailed convergence analysis of our proposed algorithm shows that the average squared norm of the objective function gradient decreases at a rate of $\mathcal{O}(1/K)$, where $K$ is the number of iterations, recovering the convergence rate of state-of-the art FMTL (Smith et al., 2017; Dinh et al., 2022).

- Extensive simulation results demonstrate the performance of our proposed algorithms on several benchmark datasets compared to state-of-the-art approaches.

## 2 RELATED WORK

**Federated Learning (FL).** FL is designed to train models on decentralized user data without sharing raw data. While numerous FL algorithms (McMahan et al., 2017; Karimireddy et al., 2020; Li et al., 2020a; Lin et al., 2020; Elgabli et al., 2022) have been proposed, most of them typically follow a similar iterative procedure where a server sends a global model to clients for updates. Then, each client trains a local model using its data and sends it back to the server for aggregation to update the global model. However, due to significant variability in locally collected data across clients, data heterogeneity poses a serious challenge. A prevalent assumption within FL literature is that the model size is the same across clients. However, recent works (Zhang et al., 2024; Liu et al., 2022b) have highlighted the significance of incorporating heterogeneous model sizes in FL frameworks in the presence of a parameter server.

**Personalized FL (PFL).** To address the challenges arising from data heterogeneity, PFL aims to learn individual client models through collaborative training, using different techniques such as local fine-tuning (Wang et al., 2019; Yu et al., 2020), meta-learning (Fallah et al., 2020; Chen et al., 2018; Jiang et al., 2019), layer personalization (Arivazhagan et al., 2019; Liang et al., 2020; Collins et al., 2021), model mixing (Hanzely & Richtárik, 2020; Deng et al., 2020a), and model-parameter regularization (T Dinh et al., 2020; Li et al., 2021; Huang et al., 2021; Liu et al., 2022a). One way to personalize FL is to learn a global model and then fine-tune its parameters at each client using a few stochastic gradient descent steps, as in (Yu et al., 2020). Per-FedAvg (Fallah et al., 2020) combines meta-learning with FedAvg to produce a better initial model for each client. Algorithms such as FedPer (Arivazhagan et al., 2019), LG-FedAvg (Liang et al., 2020), and FedRep (Collins et al., 2021) involve layer-based personalization, where clients share certain layers while training personalized layers locally. A model mixing framework for PFL, where clients learn a mixture of the global model and local models was proposed in (Hanzely & Richtárik, 2020). In (T Dinh et al., 2020), pFedMe uses an $L_2$ regularization to restrict the difference between the local and global parameters.

**Heterogeneous FL (HFL).** Unlike traditional FL paradigms that assume uniform model structures across all clients, HFL accommodates variations in model sizes, layers, and computational capabilities, thereby enabling more flexible and inclusive participation. One of the primary challenges in HFL is ensuring effective knowledge sharing and aggregation among clients with disparate models. To tackle this, many HFL approaches leverage knowledge distillation techniques, where a public dataset or a subset of data is used to distill knowledge from heterogeneous local models into a unified global model (Li & Wang, 2019; Zhu et al., 2021). For instance, FedMD (Li & Wang, 2019) employs model distillation to allow clients with different model architectures to contribute to a shared global model without necessitating architectural alignment. Additionally, techniques such as adaptive model aggregation (Zhai et al., 2024) and model compatibility layers (Setayesh et al., 2023) have been proposed to facilitate the seamless integration of diverse model updates. Another significant aspect of HFL is the consideration of clients' varying computational resources and communication capabilities. Moreover, recent advancements have introduced the use of parameter-efficient fine-tuning methods to support heterogeneity in model architectures (Chen et al., 2024).

**Federated Multi-Task Learning (FMTL).** FMTL aims to train separate but related models simultaneously across multiple clients, each potentially focusing on different but related tasks. It can be viewed as a form of PFL by considering the process of learning one local model as a single task. Multi-task learning was first introduced into FL in (Smith et al., 2017). The authors proposed MOCHA, an FMTL algorithm that jointly learns the local models as well as a task relationship matrix, which captures the relations between tasks. In the context of FMTL, task similarity can be represented through graphs, matrices, or clustering. In (Sattler et al., 2020), clustered FL, an FL framework that groups participating clients based on their local data distribution was proposed. The proposed method tackles the issue of heterogeneity in the local datasets by clustering the clients with similar data distributions and training a personalized model for each cluster. FedU, an FMTL algorithm that encourages model parameter proximity for similar tasks via Laplacian regularization, was introduced in (Dinh et al., 2022). In (SarcheshmehPour et al., 2023), the authors leverage a

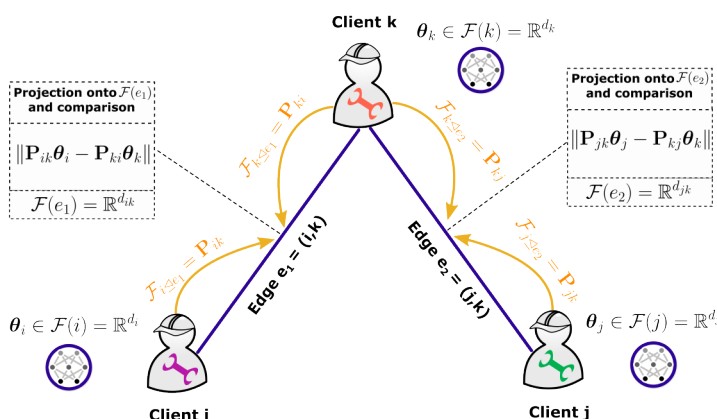

Figure 1: Schematic illustration of the sheaf-based modeling of FMTL.

generalized total variation minimization approach to cluster the local datasets and train the local models for decentralized collections of datasets with an emphasis on clustered FL. A more in-depth comparison between PFL, HFL, and FMTL can be found in Appendix C.

**Sheaves.** A major limitation in FMTL over a graph, e.g., FMTL with graph Laplacian regularization in (Dinh et al., 2022) and FMTL with generalized total variance minimization in (SarcheshmehPour et al., 2023), that we wish to address in this work, is their inability to deal with feature heterogeneity between clients. In contrast, *sheaves*, a well-established notion in algebraic topology, can inherently model higher-order relationships among heterogeneous clients. Despite the limited appearance of sheaves in the engineering domain, their importance in organizing information/data distributed over multiple clients/systems has been emphasized in the recent literature (Robinson, 2014; 2013; Riess & Ghrist, 2022). In fact, sheaves can be considered as the canonical data structure to systematically organize local information so that useful global information can be extracted (Robinson, 2017). The above-mentioned graph models with node features lying in some fixed space can be considered as the simplest examples of sheaves, where such a graph is equivalent to a *constant sheaf* structure that directly follows from the graph. Motivated by these ideas, our work focuses on using the generality of sheaves to propose a generic framework for FMTL with both data and feature heterogeneity over nodes, generalizing the works of (Dinh et al., 2022; SarcheshmehPour et al., 2023). The analogous generalization of the graph Laplacian in the sheaf context is the *sheaf Laplacian*. In the context of distributed optimization, (Hansen & Ghrist, 2019) consider sheaf Laplacian regularization with sheaf constraints, i.e., *Homological Constraints*, and the resulting saddle-point dynamics.

## 3 SHEAF-BASED FEDERATED MULTI-TASK LEARNING (SHEAF−FMTL)

### 3.1 FMTL PROBLEM SETTING

We consider a connected network of $N$ clients modeled by a graph $\mathcal{G} = (\mathcal{V}, \mathcal{E})$, where $\mathcal{V} = [N] = \{1, \ldots, N\}$ is the set of clients, and $\mathcal{E} \subseteq \mathcal{V} \times \mathcal{V}$ represents the set of edges, i.e., the set of pairs of clients that can communicate with each other. Each client $i \in \mathcal{V}$ has a local loss function $f_i : \mathbb{R}^{d_i} \to \mathbb{R}$ and only has access to its own local data distribution $\mathcal{D}_i$. Client $i$ can only communicate with the set of its neighbors defined as $\mathcal{N}_i = \{j \in \mathcal{V} | (i, j) \in \mathcal{E}\}$ whose cardinality is $|\mathcal{N}_i| = \delta_i$. In this work, we aim to fit different models, i.e., $\boldsymbol{\theta}_i \in \mathbb{R}^{d_i}, \forall i \in [N]$, to the local data of clients, while accounting for the interactions between these models. Finally, let $\boldsymbol{\theta} = [\boldsymbol{\theta}_1^T, \ldots, \boldsymbol{\theta}_N^T]^T \in \mathbb{R}^d$ be the stack of the local decision variables $\{\boldsymbol{\theta}_i\}_{i=1}^N$, where $d = \sum_{i=1}^N d_i$.

### 3.2 A SHEAF THEORETIC APPROACH OF THE FMTL PROBLEM

A "cellular sheaf" $\mathcal{F}$ of $\mathbb{R}$-vector spaces over a simple graph $\mathcal{G} = (\mathcal{V}, \mathcal{E})$, i.e., without loops and multiple edges, consists of the following assignments

- For each $i \in \mathcal{V}$, a vector space $\mathcal{F}(i) = \mathbb{R}^{d_i}$ of dimension $d_i$,

- for each edge $e = (i, j) \in \mathcal{E}$, a vector space $\mathcal{F}(e) = \mathbb{R}^{d_{ij}}$ of dimension $d_{ij}$, and

- for each edge $e \in \mathcal{E}$ and a vertex $i \in \mathcal{V}$ that is incident to the edge $e$, a linear transformation $\mathcal{F}_{i \trianglelefteq e} : \mathcal{F}(i) \to \mathcal{F}(e)$.

We shall refer to $\mathcal{F}(i)$ and $\mathcal{F}(e)$ as stalks over $i$ and $e$, respectively, and the map $\mathcal{F}_{i \trianglelefteq e}$ as the restriction map from $i$ to $e$. Also, given an edge $e = (i, j) \in \mathcal{E}$, we denote the matrix representation of $\mathcal{F}_{i \trianglelefteq e}$, with respect to a chosen basis such as the standard basis, by $\boldsymbol{P}_{ij}$. With an abuse of notation, we use $\mathcal{F}_{i \trianglelefteq e}$ and $\boldsymbol{P}_{ij}$ interchangeably, as required, in the remainder of the paper.

Naturally associated with such a sheaf structure are the dual maps, $\mathcal{F}_{i \trianglelefteq e}^* : \mathcal{F}(e) \to \mathcal{F}(i)$, of the restriction maps. Note that we are using the identification of the dual of a finite-dimensional vector space with itself. It is a standard fact that the matrix representation of the dual map $\mathcal{F}_{i \trianglelefteq e}^*$ is given by the transpose of $\boldsymbol{P}_{ij}$. Similarly, with an abuse of notation, we shall also use $\mathcal{F}_{i \trianglelefteq e}^*$ and $\boldsymbol{P}_{ij}^T$ interchangeably.

For each $i \in \mathcal{V}$, $\mathcal{F}(i)$ is the space in which the local model of client $i$ is parameterized, i.e., $\boldsymbol{\theta}_i \in \mathcal{F}(i)$. A choice $\{\boldsymbol{\theta}_i\}_{i \in \mathcal{V}}$ of local models lies in the total space $C^0(\mathcal{F}) := \bigoplus_{i \in \mathcal{V}} \mathcal{F}(i)$. Note that, we do not assume $d_i$ and $d_j$ to be the same for $i \neq j$. In particular, different clients can have different model sizes that could arise from feature heterogeneity and/or different learning tasks. Also note that an element of $C^0(\mathcal{F})$ is not fully observable by a single client, as assumed in the FL setting.

Therefore, any two models can be compared via the restriction maps, provided they share an edge. More specifically, as can be seen from Figure 1, for two clients $i$ and $j$ such that $e = (i, j) \in \mathcal{E}$, $\mathcal{F}(e)$ can be considered as the "disclose space" in which models $\boldsymbol{\theta}_i$ and $\boldsymbol{\theta}_j$ are compared via the projections $\mathcal{F}_{i \trianglelefteq e}(\boldsymbol{\theta}_i) = \boldsymbol{P}_{ij}$ and $\mathcal{F}_{j \trianglelefteq e}(\boldsymbol{\theta}_j) = \boldsymbol{P}_{ji}$. Here, the local models $\boldsymbol{\theta}_i$ are assigned to the vertices (the clients), while the restriction maps $\boldsymbol{P}_{ij}$ project the local models onto the edge space capturing the shared features or relationships between the clients.

We refer the reader to Appendix B for an interpretation of this viewpoint in the context of linear models. Given a choice of local models, the overall comparison of these models is done in the total space $C^1(\mathcal{F}) := \bigoplus_{e \in \mathcal{E}} \mathcal{F}(e)$ of the disclose spaces. The total discrepancy of such a choice of local models, as measured by comparing their projections onto the disclose spaces, can be formalized via the Laplacian quadratic form associated with the so-called "sheaf Laplacian", the analogous to the graph Laplacian. To define the Laplacian in the sheaf setting, we first need to orient the edges and define the "co-boundary map" $\delta : C^0(\mathcal{F}) \to C^1(\mathcal{F})$.

From now onwards, we shall fix an orientation for each edge and write $e = (i, j)$ for an oriented edge. For such an oriented edge $e = (i, j)$, write $e^+ = j$ and $e^- = i$. Our discussion is not subjective to the choice of orientation; however, one can choose a canonical orientation associated with an ordering of the vertices, e.g., when vertices are indexed by numbers, by choosing $e^+ = \max\{i, j\}$ and $e^- = \min\{i, j\}$ for an unoriented edge $e = \{i, j\}$. Given such an orientation, the co-boundary map $\delta$ is given as follows. For $\boldsymbol{\theta} = (\boldsymbol{\theta}_i)_{i \in \mathcal{V}}$, the co-boundary of $\boldsymbol{\theta}$, $\delta(\boldsymbol{\theta}) = (\delta(\boldsymbol{\theta})_e)_{e \in \mathcal{E}} \in C^1(\mathcal{F})$, is defined by

$$\delta(\boldsymbol{\theta})_e = \mathcal{F}_{e^+ \trianglelefteq e}(\boldsymbol{\theta}_{e^+}) - \mathcal{F}_{e^- \trianglelefteq e}(\boldsymbol{\theta}_{e^-}). \tag{1}$$

As in the case of restriction maps, one also has the dual $\delta^* : C^1(\mathcal{F}) \to C^0(\mathcal{F})$ of the co-boundary map $\delta$. The sheaf Laplacian $L_{\mathcal{F}} : C^0(\mathcal{F}) \to C^0(\mathcal{F})$ associated with the cellular sheaf $\mathcal{F}$ is then given by $L_{\mathcal{F}} = \delta^* \circ \delta$. For a given $\boldsymbol{\theta}$, the sheaf Laplacian is defined as $L_{\mathcal{F}}(\boldsymbol{\theta}) = \left( L_{\mathcal{F}}(\boldsymbol{\theta})_j \right)_{j \in \mathcal{V}}$, where $L_{\mathcal{F}}(\boldsymbol{\theta})_j = \sum_{i \in \mathcal{V}} L_{j,i}(\boldsymbol{\theta}_i)$ is given by

$$L_{j,i} = \begin{cases} \sum_{i \trianglelefteq e} \mathcal{F}_{i \trianglelefteq e}^* \circ \mathcal{F}_{i \trianglelefteq e}, & \text{if } i = j, \\ -\mathcal{F}_{j \trianglelefteq e}^* \circ \mathcal{F}_{i \trianglelefteq e}, & \text{if } e = (i, j) \in \mathcal{E}, \\ \boldsymbol{0}, & \text{otherwise.} \end{cases} \tag{2}$$

In particular, based on the ordering of $\mathcal{V}$ that is used to stack the local models, and the chosen bases for $\mathcal{F}(i)$'s and $\mathcal{F}(e)$'s, $L_{\mathcal{F}}$ is a block matrix structure indexed by $\mathcal{V}$, whose $(j, i)$ block can be directly obtained from (2). More specifically, with an abuse of notation, writing $L_{j,i}$ in terms of its

matrix representation, we have that

$$L_{j,i} = \begin{cases} \sum_{j' \in \mathcal{N}_i} \boldsymbol{P}_{ij'}^T \boldsymbol{P}_{ij'}, & \text{if } i = j, \\ -\boldsymbol{P}_{ji}^T \boldsymbol{P}_{ij}, & \text{if } e = (i,j) \in \mathcal{E}, \\ \boldsymbol{0}, & \text{otherwise.} \end{cases} \tag{3}$$

In fact, one can write the matrix representation of the co-boundary map, $\boldsymbol{P}$, as follows. It has a block structure whose rows are indexed by edges and columns are indexed by vertices. Then, the submatrix $\boldsymbol{P}_{e,i}$ that corresponds to row $e \in \mathcal{E}$ and column $i \in \mathcal{V}$ is given by

$$\boldsymbol{P}_{e,i} = \begin{cases} \boldsymbol{P}_{ij}, & \text{if } e = (i,j) \\ -\boldsymbol{P}_{ij}, & \text{if } e = (j,i), \\ \boldsymbol{0}, & \text{otherwise.} \end{cases} \tag{4}$$

From the definition $L_{\mathcal{F}} = \delta^* \circ \delta$, one can get the matrix form of $L_{\mathcal{F}}$ as $L_{\mathcal{F}} = \boldsymbol{P}^T \boldsymbol{P}$. Assuming the matrix representation of $L_{\mathcal{F}}$, we often write $L_{\mathcal{F}}(\boldsymbol{\theta}) = L_{\mathcal{F}} \boldsymbol{\theta} = \boldsymbol{P}^T \boldsymbol{P} \boldsymbol{\theta}$. Note that this block structure aligns well with the distributed optimization goal of the FMTL setting.

The significance of the sheaf Laplacian $L_{\mathcal{F}}$ is characterized by the consensus property given by

$$\ker(L_{\mathcal{F}}) = \left\{ (\boldsymbol{\theta}_i)_{i \in \mathcal{V}} \in C^0(\mathcal{F}) \,|\, \mathcal{F}_{i \trianglelefteq e}(\boldsymbol{\theta}_i) = \mathcal{F}_{j \trianglelefteq e}(\boldsymbol{\theta}_j) \text{ for } e = (i,j) \in \mathcal{E} \right\}. \tag{5}$$

In other words, $\ker(L_{\mathcal{F}})$ consists of the choices of local models that are in *global consensus* so that any two comparable local models $\boldsymbol{\theta}_i$ and $\boldsymbol{\theta}_j$ agree when projected onto the disclose space $\mathcal{F}(e)$, where $e = (i,j) \in \mathcal{E}$. Accordingly, the global consensus constraint on $\boldsymbol{\theta} \in C^0(\mathcal{F})$ is given by $L_{\mathcal{F}} \boldsymbol{\theta} = \boldsymbol{0}$.

Similar to that of a graph Laplacian, the sheaf Laplacian quadratic form $Q_{\mathcal{F}}(\boldsymbol{\theta}) = \boldsymbol{\theta}^T L_{\mathcal{F}} \boldsymbol{\theta}$ quantifies by how much a given $\boldsymbol{\theta}$ deviates from the constraint $L_{\mathcal{F}} \boldsymbol{\theta} = \boldsymbol{0}$. The following lemma shows that the sheaf Laplacian quadratic form measures the total discrepancy between the projections of the local models onto the edge spaces, summed over all edges in the graph.

**Lemma 3.1.**

$$\boldsymbol{\theta}^T L_{\mathcal{F}} \boldsymbol{\theta} = \sum_{e=(i,j) \in \mathcal{E}} \| \mathcal{F}_{i \trianglelefteq e}(\boldsymbol{\theta}_i) - \mathcal{F}_{j \trianglelefteq e}(\boldsymbol{\theta}_j) \|^2 = \sum_{e=(i,j) \in \mathcal{E}} \| \boldsymbol{P}_{ij} \boldsymbol{\theta}_i - \boldsymbol{P}_{ji} \boldsymbol{\theta}_j \|^2. \tag{6}$$

*Proof.* The details of the proof are deferred to Appendix G. $\square$

Next, we show that $\boldsymbol{\theta}$ being a global section, i.e., $\boldsymbol{\theta} \in \ker(L_{\mathcal{F}})$, is equivalent to $\boldsymbol{\theta}$ minimizing the sheaf Laplacian quadratic form.

**Lemma 3.2.**

$$\ker(L_{\mathcal{F}}) = \underset{\boldsymbol{\theta} \in C^0(\mathcal{F})}{\arg\min} Q_{\mathcal{F}}(\boldsymbol{\theta}) = \underset{\boldsymbol{\theta} \in C^0(\mathcal{F})}{\arg\min} \boldsymbol{\theta}^T L_{\mathcal{F}} \boldsymbol{\theta}. \tag{7}$$

*Proof.* The proof is provided in Appendix G. $\square$

In the context of FMTL, a cellular sheaf is a structured way to assign information to each client (node) and their connections (edges) in a network as follows

- Nodes (Clients): Each client $i$ has its own local model $\boldsymbol{\theta}_i \in \mathbb{R}^{d_i}$.

- Edges (Interaction space): Each connection between clients $i$ and $j$ has a shared space $\mathcal{F}(e) \in \mathbb{R}^{d_{ij}}$.

- Restriction Maps: The mappings $\boldsymbol{P}_{ij} : \mathbb{R}^{d_i} \to \mathbb{R}^{d_{ij}}$ project the local model of client $i$ into the shared space with client $j$. This projection facilitates meaningful comparisons and collaborations between clients, ensuring that heterogeneous models can still interact effectively within the network.

This setup allows us to systematically ensure that the models of connected clients align within their shared interaction spaces, promoting consistency and cooperation across the network. To formulate the FMTL optimization problem, we aim to minimize the combined objectives of individual client losses and a regularization term that enforces consistency across client models. Specifically, each client seeks to minimize its own loss function based on its local data, while the regularization term penalizes discrepancies between connected clients in their shared interaction spaces. Building upon this, we can express the regularization term more succinctly using the sheaf Laplacian matrix $L_{\mathcal{F}}(\boldsymbol{P})$. This matrix encapsulates the structural relationships and shared interaction spaces between clients, allowing us to reformulate the optimization problem in a compact and mathematically elegant manner as follows

$$\min_{\boldsymbol{\theta}, \boldsymbol{P}} \Psi(\boldsymbol{\theta}, \boldsymbol{P}) = f(\boldsymbol{\theta}) + \frac{\lambda}{2} \boldsymbol{\theta}^T L_{\mathcal{F}}(\boldsymbol{P}) \boldsymbol{\theta}, \tag{8}$$

where $f(\boldsymbol{\theta}) = \sum_{i=1}^{N} f_i(\boldsymbol{\theta}_i)$ and $L_{\mathcal{F}}(\boldsymbol{P})$ is the sheaf Laplacian for the choices of the restriction maps. The sheaf Laplacian regularization term $\frac{\lambda}{2} \boldsymbol{\theta}^T L_{\mathcal{F}}(\boldsymbol{P}) \boldsymbol{\theta}$ enforces consistency between the projections of the local models onto the edge space, promoting collaboration among the clients. A more in-depth discussion on the rationale behind using Sheaf theory to model clients' interaction in the context of FMTL can be found in Appendix A. In Appendix D, we show that our proposed framework is very general and covers many special cases previously introduced in the literature.

### 3.3 PROPOSED ALGORITHM & CONVERGENCE ANALYSIS

Note that problem (8) arising from the sheaf formulation can be re-written as follows

$$\min_{\substack{\{\boldsymbol{\theta}_i\}_{i \in \mathcal{V}}, \\ \{\boldsymbol{P}_{ij}\}_{(i,j) \in \mathcal{E}}}} \sum_{i=1}^{N} f_i(\boldsymbol{\theta}_i) + \frac{\lambda}{2} \sum_{i=1}^{N} \sum_{j \in \mathcal{N}_i} \|\boldsymbol{P}_{ij} \boldsymbol{\theta}_i - \boldsymbol{P}_{ji} \boldsymbol{\theta}_j\|^2, \tag{9}$$

where $\|\cdot\|$ is the Euclidean norm, and $\forall (i,j) \in \mathcal{E}$, $\boldsymbol{P}_{ij}$ is a matrix with size $(d_{ij}, d_i)$ such that $d_{ij} = d_{ji}$. In (9), we propose to jointly learn the models $\{\boldsymbol{\theta}_i\}_{i \in \mathcal{V}}$ and the matrices $\{\boldsymbol{P}_{ij}\}_{(i,j) \in \mathcal{E}}$. The matrix $\boldsymbol{P}_{ij}$ can be seen as an encoding or compression matrix since it maps the higher-dimensional vector $\boldsymbol{\theta}_i$ to a lower-dimensional space with dimension $d_{ij}$, effectively retaining only the most important features or information shared between the two clients $i$ and $j$. Hence, the term $(\boldsymbol{P}_{ij} \boldsymbol{\theta}_i - \boldsymbol{P}_{ji} \boldsymbol{\theta}_j)$ captures the dissimilarity or discrepancy between the two vectors $\boldsymbol{\theta}_i$ and $\boldsymbol{\theta}_j$ in this shared subspace.

**Remark 3.3.** *In (9), the hyperparameter $\lambda$ controls the impact of the models of neighboring clients on each local model. When $\lambda > 0$, the minimization of the regularization term promotes the proximity among the models of neighboring clients. On the other hand, if $\lambda = 0$, (9) reduces to an individual learning problem, wherein each client independently learns its local model $\boldsymbol{\theta}_i$ solely from its local data, without engaging in any collaborative efforts with the other clients. Finally, as $\lambda \to \infty$, (9) boils down to the classical FL problem where the aim is to learn a global model (Hanzely & Richtárik, 2020).*

Next, we propose a communication-efficient algorithm to solve (9) by adopting an iterative optimization approach. Since the objective function is assumed to be differentiable, we can use gradient-based optimization methods. More specifically, we will use alternating gradient descent (AGD) updates for $\{\boldsymbol{\theta}_i\}_{i \in [N]}$ and $\{\boldsymbol{P}_{ij}\}_{(i,j) \in \mathcal{E}}$, respectively. At iteration $(k+1)$, client $i$ sends $\boldsymbol{P}_{ij}^k \boldsymbol{\theta}_i^k$ and receives $\{\boldsymbol{P}_{ji}^k \boldsymbol{\theta}_j^k\}_{j \in \mathcal{N}_i}$ from its neighbours, to update its model $\boldsymbol{\theta}_i$, using one gradient descent step

$$\boldsymbol{\theta}_i^{k+1} = \boldsymbol{\theta}_i^k - \alpha \left( \nabla f_i(\boldsymbol{\theta}_i^k) + \lambda \sum_{j \in \mathcal{N}_i} (\boldsymbol{P}_{ij}^k)^T (\boldsymbol{P}_{ij}^k \boldsymbol{\theta}_i^k - \boldsymbol{P}_{ji}^k \boldsymbol{\theta}_j^k) \right). \tag{10}$$

Then, client $i$ sends $\boldsymbol{P}_{ij}^k \boldsymbol{\theta}_i^{k+1}$ and receives $\{\boldsymbol{P}_{ji}^k \boldsymbol{\theta}_j^{k+1}\}_{j \in \mathcal{N}_i}$ from its neighbours, to be able to update its matrices $\{\boldsymbol{P}_{ij}\}_{j \in \mathcal{N}_i}$, using one gradient descent step, according to

$$\boldsymbol{P}_{ij}^{k+1} = \boldsymbol{P}_{ij}^k - \eta \lambda (\boldsymbol{P}_{ij}^k \boldsymbol{\theta}_i^{k+1} - \boldsymbol{P}_{ji}^k \boldsymbol{\theta}_j^{k+1})(\boldsymbol{\theta}_i^{k+1})^T, \tag{11}$$

where $\alpha$ and $\eta$ are two learning rates. Note that, in our proposed algorithm, neighbouring clients only share vectors and no matrix exchange is needed in both updates (10) and (11). Our proposed method is summarized in Algorithm 1.

---

**Algorithm 1** Sheaf-based Federated Multi-Task Learning (`Sheaf-FMTL`)

---

**Parameters:** number of clients $N$, number of iterations $K$, learning rates $(\alpha, \eta)$, regularization parameter $\lambda$.

**Initialization:** initial models $\{\boldsymbol{\theta}_i^0\}_{i=1}^N$, initial matrices $\{\boldsymbol{P}_{ij}^0\}_{(i,j)\in\mathcal{E}}$.

**for** $k = 0, \ldots, K$ **do**

    **for client** $i = 1, \ldots, N$ in parallel **do**

        ▷ Sends $\boldsymbol{P}_{ij}^k \boldsymbol{\theta}_i^k$ and receives $\boldsymbol{P}_{ji}^k \boldsymbol{\theta}_j^k$ from each neighbour $j \in \mathcal{N}_i$

        ▷ Updates its model

$$\boldsymbol{\theta}_i^{k+1} = \boldsymbol{\theta}_i^k - \alpha \left( \nabla f_i(\boldsymbol{\theta}_i^k) + \lambda \sum_{j \in \mathcal{N}_i} (\boldsymbol{P}_{ij}^k)^T (\boldsymbol{P}_{ij}^k \boldsymbol{\theta}_i^k - \boldsymbol{P}_{ji}^k \boldsymbol{\theta}_j^k) \right).$$

        ▷ Sends $\boldsymbol{P}_{ij}^k \boldsymbol{\theta}_i^{k+1}$ and receives $\boldsymbol{P}_{ji}^k \boldsymbol{\theta}_j^{k+1}$ from each neighbour $j \in \mathcal{N}_i$

        ▷ Updates its matrix

$$\boldsymbol{P}_{ij}^{k+1} = \boldsymbol{P}_{ij}^k - \eta\lambda(\boldsymbol{P}_{ij}^k \boldsymbol{\theta}_i^{k+1} - \boldsymbol{P}_{ji}^k \boldsymbol{\theta}_j^{k+1})(\boldsymbol{\theta}_i^{k+1})^T.$$

    **end for**

**end for**

---

**Remark 3.4.** *Note that each neighbour of the node $i$ is required to send the vector $\boldsymbol{P}_{ji}\boldsymbol{\theta}_j$ in order to update $\boldsymbol{\theta}_i$ and $\boldsymbol{P}_{ij}$ as per (10) and (11). The dimension of $\boldsymbol{P}_{ij}\boldsymbol{\theta}_j$ is $d_{ij}$, which in practice could be much smaller than $d_j$, the size of $\boldsymbol{\theta}_j$. For example, a reasonable choice of $d_{ij}$ is $d_{ij} = \min(d_i, d_j)$. Hence, our proposed algorithm is more communication-efficient than sending the models $\{\boldsymbol{\theta}_i\}_{i\in\mathcal{V}}$.*

Next, we turn to analyzing the convergence of `Sheaf-FMTL`. To this end, we start by making the following standard assumptions.

**Assumption 1 (Smoothness).** $\forall i \in [N]$, the function $f_i$ is assumed to be $L$-smooth, i.e., there exists $L > 0$ such that $\forall i \in [N], \forall \boldsymbol{\theta}_1, \boldsymbol{\theta}_2, \|\nabla f_i(\boldsymbol{\theta}_2) - \nabla f_i(\boldsymbol{\theta}_1)\| \leq L\|\boldsymbol{\theta}_2 - \boldsymbol{\theta}_1\|$.

**Assumption 2 (Bounded domain).** There exists $D_\theta > 0$ such that $\|\boldsymbol{\theta}\| \leq D_\theta$.

Assumptions 1-2 are key assumptions that are often used in the context of distributed optimization (Karimireddy et al., 2020; Li et al., 2020a; Deng et al., 2020b; 2023). In particular, Assumption 2 is commonly used in the convex-concave minimax literature (Deng et al., 2020b; 2023). The following theorem establishes the convergence rate of the `Sheaf-FMTL` algorithm.

**Theorem 3.5.** *Let Assumptions 1 and 2 hold, and the learning rates $\alpha$ and $\eta$ satisfy the conditions $\alpha < \frac{2}{NL}$ and $\eta < \frac{2}{\lambda D_\theta^2}$, respectively. Then, the averaged gradient norm is upper bounded as follows*

$$\frac{1}{K}\sum_{k=0}^{K-1} \|\nabla\Psi(\boldsymbol{\theta}^k, \boldsymbol{P}^k)\|^2 \leq \frac{1}{\rho K}(\Psi(\boldsymbol{\theta}^0, \boldsymbol{P}^0) - \Psi^\star), \tag{12}$$

*where $\rho = \min\left\{\alpha\left(1 - \frac{\alpha NL}{2}\right), \eta\left(1 - \frac{\eta\lambda D_\theta^2}{2}\right)\right\}$ and $\Psi^\star$ is the optimal value of $\Psi$.*

*Proof.* The proof can be found in Appendix H. $\qquad\square$

## 4 EXPERIMENTS

### 4.1 EXPERIMENTAL SETUP

To validate our theoretical foundations, we numerically evaluate the performance of our proposed algorithm `Sheaf-FMTL` using two experiments: (i) the clients have the same model size in Section 4.2, and (ii) the clients have different model sizes in Section 4.3.

**Datasets.** In the first experiment, we examine two datasets: Rotated MNIST and Heterogeneous CIFAR-10. A detailed description of the datasets can be found in Appendix I.1. In appendix J,

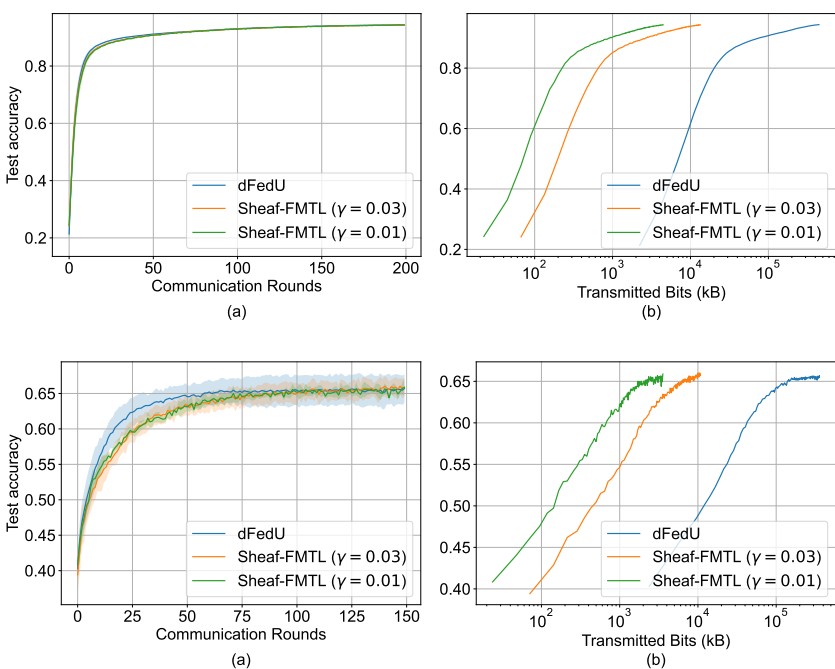

Figure 2: Test accuracy as a function of the number of communication rounds and the number of transmitted bits for the Rotated MNIST dataset (top), and the Heterogenous CIFAR-10 dataset (bottom).

we report additional results using four more datasets. In the second experiment, we used modified versions of the Vehicle and School datasets by randomly dropping features of the local datasets to make the model size different across clients. More details on the generation of these two datasets can be found in Appendix I.2.

**Baselines.** In the first experiment, we compare to the dFedU algorithm (Dinh et al., 2022). We implement our proposed algorithm using a mini-batch stochastic gradient in (10) to make the comparison fair. In the second one, we compare to a stand-alone baseline where each client trains on each local dataset without communicating with the rest of the clients. To the best of our knowledge, `Sheaf-FMTL` is the only algorithm solving the FMTL problem over decentralized topology with the clients having different model sizes, hence the comparison to a stand-alone baseline in the second experiment. More details on the experimental Settings can be found in Appendix I.2.

### 4.2 EXPERIMENT 1: SAME MODEL SIZE

Figure 2 illustrates the performance of the proposed `Sheaf-FMTL` algorithm and dFedU on the Rotated MNIST and Heterogeneous CIFAR-10 datasets, respectively, showcasing the test accuracy as a function of the number of communication rounds and the total number of transmitted bits. We consider two values for $\gamma = \{0.01, 0.03\}$ such that the projection space dimension is $\gamma d$. In Figure 2(a), we can see that `Sheaf-FMTL` manages to achieve similar test accuracies as dFedU. On the other hand, Figure 2(b) shows that `Sheaf-FMTL` achieves higher test accuracy with fewer transmitted bits compared to the baseline, demonstrating its ability to learn effectively while minimizing communication overhead. For the Rotated MNIST dataset, using $\gamma = 0.01$ leads to almost similar test accuracy as dFedU, while it requires $100\times$ less in terms of the number of transmitted bits to achieve this accuracy. For the Heterogeneous CIFAR-10, `Sheaf-FMTL` requires slightly more communication rounds than dFedU but the benefit in terms of communication overhead is evident as it requires exchanging significantly less number of bits.

As illustrated in Table 1, `Sheaf-FMTL` incurs additional storage and computational costs due to the maintenance and training of restriction maps. Specifically, each restriction map requires storing

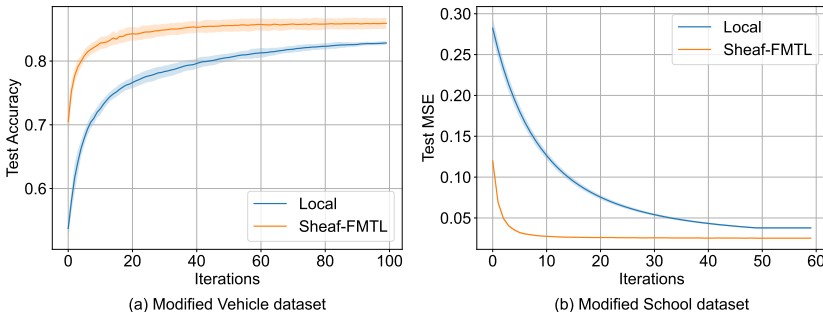

(a) Modified Vehicle dataset            (b) Modified School dataset

Figure 3: Test accuracy/MSE as a function of the number of iterations for (a) Modified Vehicle dataset, and (b) Modified School dataset.

Table 1: Comparative analysis of storage, computational overheads, and communication costs.

| Method | Storage per client | Compute per iteration per client | Communication per iteration per client |
|---|---|---|---|
| Sheaf-FMTL | $d_i + \sum_{j \in \mathcal{N}_i} d_{ij} \times d_i$ | $\mathcal{O}(d_i \times d_{ij})$ | $\sum_{j \in \mathcal{N}_i} d_{ij}$ |
| dFedU | $d_i$ | $\mathcal{O}(d_i)$ | $\sum_{j \in \mathcal{N}_i} d_i$ |

a matrix of size $d_{ij} \times d_i$, leading to a cumulative storage requirement of $\mathcal{O}(|\mathcal{E}| \times d_{ij} \times d_i)$ across the network. Computationally, updating these maps involves matrix multiplications and gradient calculations, adding a complexity of $\mathcal{O}(d_{ij} \times d_i)$ per edge per iteration. However, these costs are significantly offset by the substantial communication savings achieved, particularly when $d_{ij}$ is chosen to be a small fraction of $d_i$, making Sheaf-FMTL a viable option in resource-rich FL environments such as cross-silo FL settings. For a more detailed analysis of the scalability of Sheaf-FMTL with more complex data and larger parameter spaces, including neural networks, as well as strategies to mitigate computational and storage overheads, please refer to Appendix E.

### 4.3 EXPERIMENT 2: DIFFERENT MODEL SIZES

Figure 3 compares the performance of the proposed Sheaf-FMTL algorithm with the local training, i.e., training each model independently without communication with other clients, on two modified versions of the Vehicle and School datasets by plotting the test accuracy as a function of the number of iterations. We can see that Sheaf-FMTL demonstrates a clear advantage over the local training for both datasets. This observation highlights the effectiveness of Sheaf-FMTL in leveraging the shared information across clients while preserving the individual characteristics of their local models, a key feature of FL. By utilizing the sheaf structure to capture the relationships between the local models, Sheaf-FMTL enables more efficient and accurate learning than the local approach when the model size is different across clients.

## 5 CONCLUSION

In this work, we introduced a novel sheaf-based framework for federated multi-task learning that effectively tackles challenges arising from data and sample heterogeneity across clients. By leveraging cellular sheaves, our approach can flexibly model complex interactions between client models, even in the presence of varying feature spaces and model sizes. The proposed Sheaf-FMTL algorithm is communication-efficient, preserves client privacy, and provides a unified view subsuming various existing FL methods. Theoretically, we analyzed the convergence properties of Sheaf-FMTL, establishing a sublinear convergence rate in line with state-of-the-art decentralized FMTL algorithms. Empirically, extensive experiments on benchmark datasets demonstrated the communication savings of Sheaf-FMTL compared to the baseline dFedU.

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

# A  SHEAF-THEORETIC APPROACH IN FMTL

## A.1  RATIONALE FOR ADOPTING SHEAF THEORY IN FMTL

Sheaf theory provides a powerful mathematical framework for modelling and analyzing complex relationships in FMTL. The adoption of this approach in our context is motivated by several key advantages

1. **Heterogeneity modeling.** FMTL often involves clients with different data distributions, model architectures, or task objectives. Sheaf theory allows us to capture these heterogeneous relationships in a structured and mathematically rigorous manner.

2. **Local-Global consistency.** Sheaves provide a natural way to ensure consistency between local (client-specific) and global (network-wide) information. This is crucial in FMTL scenarios where we aim to leverage network information to improve local performance.

3. **Flexible representation.** The sheaf structure allows for representing varying degrees of similarity or difference between clients. This nuanced representation is more sophisticated than traditional approaches that often assume uniform relationships across the network.

## A.2  THE INTERACTION SPACE AND CLIENT RELATIONSHIPS

The interaction space, denoted as $\mathcal{F}(e)$, plays a central role in our sheaf-theoretic approach to FMTL. It serves as a shared space where local models $\boldsymbol{\theta}_i$ and $\boldsymbol{\theta}_j$ are projected using restriction maps $\boldsymbol{P}_{ij}$ and $\boldsymbol{P}_{ji}$, respectively. The projection into this interaction space provides a measure of client relationships for the following reasons

1. **Common feature capture.** The interaction space captures the common or comparable features between clients, analogous to how principal component analysis (PCA) captures the most important features of a dataset.

2. **Consistency enforcement.** Our approach enforces consistency between the projections of local models onto the interaction space. This is mathematically represented by the sheaf Laplacian term $\frac{\lambda}{2}\boldsymbol{\theta}^T L_{\mathcal{F}}(\boldsymbol{P})\boldsymbol{\theta}$, which penalizes discrepancies between the projections of local models.

3. **Collaboration encouragement.** By minimizing the sheaf Laplacian term, local models are encouraged to collaborate effectively, leveraging shared information to improve overall performance.

## A.3  RESTRICTION MAPS AND THEIR INTERPRETATIONS

The restriction maps, represented by matrices $\boldsymbol{P}_{ij}$, are fundamental to our sheaf-theoretic approach. These maps project local models $\boldsymbol{\theta}_i$ onto the interaction space $\mathcal{F}(e)$. The intuition behind these maps can be understood as follows

1. **Feature selection.** $\boldsymbol{P}_{ij}$ acts as a feature selection matrix, identifying common or comparable features between clients $i$ and $j$.

2. **Information sharing.** The restriction maps facilitate information sharing between clients by projecting local models onto a common space, enabling effective collaboration even when local models have different dimensions or feature sets.

3. **Model comparison.** In heterogeneous settings where clients have different model sizes, traditional FL methods relying on direct model aggregation or comparison fail. The restriction maps allow for meaningful comparisons by projecting onto a common space.

## A.4  DIMENSIONAL CONSIDERATIONS AND TRADE-OFFS

The dimensions of the restriction map $\boldsymbol{P}_{ij}$ are determined by the dimensions of the local model $\boldsymbol{\theta}_i$ and the interaction space $\mathcal{F}(e)$. If $\boldsymbol{\theta}_i \in \mathbb{R}^{d_i}$ and the interaction space has dimension $d_{ij}$, then $\boldsymbol{P}_{ij}$ is of size $d_{ij} \times d_i$. The choice of $d_{ij}$ involves a trade-off

- **Smaller** $d_{ij}$: Results in a more compact representation of shared information, leading to communication savings.
- **Larger** $d_{ij}$: Allows for more flexibility in capturing relationships between local models but increases communication costs.

In practice, the choice of $d_{ij}$ can be guided by factors such as the estimated overlap in feature spaces between clients, computational resources available, and the desired balance between model expressiveness and communication efficiency.

### A.5 COMPARATIVE ADVANTAGES OVER TRADITIONAL FMTL METHODS

Our sheaf-theoretic approach offers several advantages over traditional FMTL methods

1. **Heterogeneity handling.** Unlike many traditional FMTL methods that assume homogeneous models across clients, our approach naturally accommodates heterogeneous model architectures and task objectives.

2. **Nuanced relationships.** Traditional methods often assume uniform relationships between clients. Our approach allows for more nuanced modelling of inter-client relationships through the interaction space and restriction maps.

3. **Privacy preservation.** By working in the interaction space rather than directly sharing model parameters, our method potentially offers enhanced privacy compared to traditional FMTL approaches.

## B INTERPRETATION OF $d_{ij}$ AND $\boldsymbol{P}_{ij}$

In this appendix, we provide an interpretation of the restriction maps $\boldsymbol{P}_{ij} = \mathcal{F}_{i \trianglelefteq e}$ for the case when the local models are given by linear or logistic regression. We describe how to choose $d_{ij}$ and $\boldsymbol{P}_{ij}$ in a meaningful way and the constraints that can be imposed on them.

Consider a network of $N$ clients, where each client $i$ has a local model parameterized by $\boldsymbol{\theta}_i \in \mathbb{R}^{d_i}$. In the case of linear regression, the local model of client $i$ is given by $f_i(\boldsymbol{\theta}_i; \boldsymbol{x}_i) = \boldsymbol{\theta}_i^T \boldsymbol{x}_i$, where $\boldsymbol{x}_i \in \mathbb{R}^{d_i}$ is the input feature vector. For logistic regression, the local model is given by $f_i(\boldsymbol{\theta}_i; \boldsymbol{x}_i) = \sigma(\boldsymbol{\theta}_i^T \boldsymbol{x}_i)$, where $\sigma(\cdot)$ is the sigmoid function.

**Interpretation of $\boldsymbol{P}_{ij}$.** Consider two clients $i, j \in \mathcal{V}$ such that $e = (i, j) \in \mathcal{E}$. The restriction maps $\boldsymbol{P}_{ij}$ and $\boldsymbol{P}_{ji}$ aim to capture the relationships between the local models $\boldsymbol{\theta}_i$ and $\boldsymbol{\theta}_j$ by projecting them to a common interaction space $\mathcal{F}(e)$. In the context of linear or logistic regression, $\boldsymbol{P}_{ij}$ and $\boldsymbol{P}_{ji}$ can be interpreted as feature selection matrices that identify the common or comparable features between the two clients.

Let $\boldsymbol{P}_{ij} \in \mathbb{R}^{d_{ij} \times d_i}$ and $\boldsymbol{P}_{ji} \in \mathbb{R}^{d_{ij} \times d_j}$ be the restriction maps for clients $i$ and $j$, respectively, where $d_{ij} = \dim \mathcal{F}(e)$ is the dimension of the interaction space. The restriction maps should satisfy the following condition

$$\boldsymbol{P}_{ij}\boldsymbol{\theta}_i \approx \boldsymbol{P}_{ji}\boldsymbol{\theta}_j, \tag{13}$$

which ensures that the projected models in the interaction space are comparable.

To impose the condition in (13), we consider the following regularizer term

$$Q_{ij} = \left\| \boldsymbol{P}_{ij}\boldsymbol{\theta}_i - \boldsymbol{P}_{ji}\boldsymbol{\theta}_j \right\|^2, \tag{14}$$

which we aim to minimize. By adding the regularizer terms for all neighboring clients and then summing over all clients, we obtain the overall regularizer

$$Q(\boldsymbol{\theta}) = \sum_{i=1}^{N} \sum_{j \in \mathcal{N}_i} \left\| \boldsymbol{P}_{ij}\boldsymbol{\theta}_i - \boldsymbol{P}_{ji}\boldsymbol{\theta}_j \right\|^2, \tag{15}$$

which is exactly the sheaf quadratic form for the choice $\mathcal{F}_{i \trianglelefteq e} = \boldsymbol{P}_{ij}$ for $e = (i, j) \in \mathcal{E}$.

**Choice of $d_{ij}$.** The dimension of the interaction space, $d_{ij}$, determines the size of the restriction

maps $\boldsymbol{P}_{ij}$ and $\boldsymbol{P}_{ji}$. In practice, $d_{ij}$ can be chosen based on the number of common or comparable features between clients $i$ and $j$. A smaller value of $d_{ij}$ implies a more compact representation of the shared information between the two clients, while a larger value allows for more flexibility in capturing the relationships between the local models.

## C DIFFERENTIATING PFL, HFL, AND FMTL

FL has evolved to encompass various paradigms that address specific challenges inherent in decentralized data environments. Among these, PFL, HFL, and FMTL stand out due to their distinct objectives and methodologies. In what follows, we elaborate on the nuances between these paradigms.

### C.1 OBJECTIVE

FMTL, PFL, and HFL each pursue distinct goals within the FL landscape. FMTL aims to enable collaborative learning across multiple related tasks, ensuring that each task benefits from shared knowledge while maintaining task-specific optimizations. In contrast, PFL focuses on optimizing a single shared model that is then personalized for each client, enhancing performance tailored to individual client needs. This can be achieved by introducing an $L_2$ regularization term that restricts the difference between the local and global parameters. HFL seeks to facilitate collaboration among clients that may have heterogeneous models, allowing the integration of these models without requiring uniformity in their structures.

### C.2 TASK DIVERSITY

FMTL supports multiple distinct yet related tasks across different clients, leveraging the similarities between tasks to improve overall learning efficiency and performance. PFL, on the other hand, operates under the assumption of a common task distributed among clients, with each client developing personalized adaptations of the shared model to better suit their specific data and requirements. HFL is designed to handle potentially heterogeneous models among clients, enabling collaboration without the necessity of explicitly modeling the relationships between clients.

### C.3 MODEL ARCHITECTURE

In terms of model architecture, FMTL is flexible in accommodating different architectures for each task or client, provided that the relationships between tasks are effectively modeled to facilitate knowledge sharing. PFL typically relies on a common base model that is augmented with personalized layers or adaptations for each client, ensuring a balance between shared knowledge and individual customization. HFL explicitly manages varying model architectures across clients by employing techniques such as knowledge distillation, which allows for the transfer of information between heterogeneous models without requiring a uniform architectural framework.

### C.4 KNOWLEDGE SHARING MECHANISM

The mechanisms for knowledge sharing vary significantly among FMTL, PFL, and HFL. FMTL leverages the relationships between tasks to enable effective knowledge transfer across related tasks, enhancing the learning process through shared insights. PFL shares a global model baseline that serves as the foundation for each client's personalized model, ensuring that the core knowledge is distributed while allowing for individual customization. HFL utilizes advanced techniques like knowledge distillation to transfer information between heterogeneous models, ensuring that valuable knowledge is shared even when clients operate with different model architectures.

## D SPECIAL CASES

In this appendix, we show how some previous works can be considered as special cases of the present work. Most of the works that we refer to, except (Zhang et al., 2024), consider the same model size for all the clients. Thus, unless otherwise stated, we assume in the rest of this subsection that all the local models are assumed to have the same size $p \in \mathbb{N}$, i.e., $\forall i \in \mathcal{V}, \boldsymbol{\theta}_i \in \mathbb{R}^p$. In particular, we

chose $\forall i \in \mathcal{V}$ and $\forall e \in \mathcal{E}$, $\mathcal{F}(i) = \mathcal{F}(e) = \mathbb{R}^p$. In Table 2, we summarize the connections of our framework to state-of-the-art FL approaches.

Table 2: Connection of the `Sheaf-FMTL` framework to state-of-the-art FL.

| FL Framework | Restriction Maps Choice |
|---|---|
| Conventional FMTL | $\boldsymbol{P}_{ij} = \sqrt{a_{ij}}\boldsymbol{I}_p$ |
| Conventional FL | $\boldsymbol{P}_{ij} = \boldsymbol{I}_p$ |
| Personalized FL | $\boldsymbol{P}_{0i} = \boldsymbol{P}_{i0} = \boldsymbol{I}_p, \boldsymbol{P}_{ij} = \boldsymbol{0}_p$ |
| Hybrid FL | $\boldsymbol{P}_{0i} = \boldsymbol{\Pi}_i, \boldsymbol{P}_{i0} = \boldsymbol{I}_{d_i}, \boldsymbol{P}_{ij} = \boldsymbol{0}_p$ |

Next, we show in detail how the choices of the sheaf maps, $\boldsymbol{P}_{ij}$, made in Table 2 recover some of the existing FL frameworks.

**Connection with conventional FMTL (Dinh et al., 2022).** In conventional FMTL, we aim to solve the problem

$$\min_{\{\boldsymbol{\theta}_i\}_{i \in \mathcal{V}}} \sum_{i=1}^{N} f_i(\boldsymbol{\theta}_i) + \frac{\lambda}{2} \sum_{i=1}^{N} \sum_{j \in \mathcal{N}_i} a_{ij} \|\boldsymbol{\theta}_i - \boldsymbol{\theta}_j\|^2, \tag{16}$$

where the weights $\{a_{ij}\}$ are assumed to be known in advance. This problem is a special case associated with the sheaf $\mathcal{F}$ arising by choosing $\mathcal{F}_{i \trianglelefteq e} = \boldsymbol{P}_{ij} = \sqrt{a_{ij}}\boldsymbol{I}_p$, where $\boldsymbol{I}_p$ is the $p \times p$ identity map/matrix. The dimension of the projection space is $d_{ij} = p$ for all edges $(i, j)$. For this particular choice of the sheaf $\mathcal{F}$, the associated Laplacian quadratic form is precisely

$$Q_{\mathcal{F}}(\boldsymbol{\theta}) = \boldsymbol{\theta}^T L_{\mathcal{F}} \boldsymbol{\theta} = \sum_{i,j \triangleleft e} a_{ij} \|\boldsymbol{\theta}_i - \boldsymbol{\theta}_j\|^2 = \sum_{(i,j) \in \mathcal{E}} a_{ij} \|\boldsymbol{\theta}_i - \boldsymbol{\theta}_j\|^2. \tag{17}$$

Replacing this into (9), we get the conventional FMTL problem (16).

**Connection with conventional FL (Ye et al., 2020).** Setting the restriction maps to be $\boldsymbol{P}_{ij} = \boldsymbol{I}_p$ and $d_{ij} = p$, $\forall(i, j) \in [N] \times [N]$. Then, the sheaf Laplacian regularization is given by

$$Q_{\mathcal{F}}(\boldsymbol{\theta}) = \sum_{(i,j) \in \mathcal{E}} \|\boldsymbol{\theta}_i - \boldsymbol{\theta}_j\|^2. \tag{18}$$

Hence, (9) reduces to

$$\min_{\{\boldsymbol{\theta}_i\}_{i \in \mathcal{V}}} \sum_{i=1}^{N} f_i(\boldsymbol{\theta}_i) + \frac{\lambda}{2} \sum_{(i,j) \in \mathcal{E}} \|\boldsymbol{\theta}_i - \boldsymbol{\theta}_j\|^2 \tag{19}$$

Taking $\lambda \to \infty$, as pointed out in Remark 3.3, one recovers the conventional FL problem

$$\min_{\{\boldsymbol{\theta}_i\}_{i \in \mathcal{V}}} \sum_{i=1}^{N} f_i(\boldsymbol{\theta}_i)$$
$$\text{s.t } \boldsymbol{\theta}_i = \boldsymbol{\theta}_j, \forall(i, j) \in \mathcal{E}. \tag{20}$$

**Connection with personalized FL (Hanzely et al., 2020).** Introducing the client 0, e.g., a server, where $f_0 \triangleq 0$ and $\boldsymbol{\theta}_0 = \bar{\boldsymbol{\theta}}$. Furthermore, let $\boldsymbol{P}_{0i} = \boldsymbol{P}_{i0} = \boldsymbol{I}_p$, $\forall i \in [N]$, and $\boldsymbol{P}_{ij} = \boldsymbol{0}_p$, $\forall(i, j) \in [N] \times [N]$, where $\boldsymbol{0}_p$ is the $p \times p$ zero map/matrix. Hence, the set of neighbours of client 0 is $\mathcal{N}_0 = [N]$, and the set of each client $i \in [N]$ is $\mathcal{N}_i = \{0\}$. We observe that this amounts to choosing the constant sheaf over the graph that connects each client to the server. Then, the associated Laplacian quadratic form can be written as

$$Q_{\mathcal{F}}(\boldsymbol{\theta}) = \sum_{i \in \mathcal{N}_0} \|\boldsymbol{P}_{0i}\boldsymbol{\theta}_0 - \boldsymbol{P}_{i0}\boldsymbol{\theta}_i\|^2 = \sum_{i=1}^{N} \|\bar{\boldsymbol{\theta}} - \boldsymbol{\theta}_i\|^2. \tag{21}$$

Therefore, (9) reduces to the personalized FL objective (Hanzely et al., 2020, Eq. (2))

$$\min_{\{\boldsymbol{\theta}_i\}_{i \in \mathcal{V}}} \sum_{i=1}^{N} f_i(\boldsymbol{\theta}_i) + \frac{\lambda}{2} \sum_{i=1}^{N} \left\| \boldsymbol{\theta}_i - \bar{\boldsymbol{\theta}} \right\|^2. \tag{22}$$

**Connection with hybrid FL (Zhang et al., 2024).** In this case, a communication framework is established between a server (with index 0) and a set of clients ($i \in [N]$), with each client connected solely to the server. The server has access to all features, while clients are constrained by their local features. Hence, for each client $i$, $d_i \leq d_0$, where $\boldsymbol{\theta}_i \in \mathbb{R}^{d_i}$ and $\boldsymbol{\theta}_0 \in \mathbb{R}^{d_0}$ are the models of client $i$ and the server, respectively. Let $\boldsymbol{\Pi}_i$ denote binary matrices that prune the server model to align with the client local model, referred to as the *selection matrices*. Given the above description, we have $\mathcal{F}(i) = \mathbb{R}^{d_i}$ and $\mathcal{F}(e) = \mathbb{R}^{d_i}$ for every $i \in [N]$ and edge of the form $e = (i, 0)$. The associated restriction maps are $\boldsymbol{P}_{0i} = \boldsymbol{\Pi}_i$ and $\boldsymbol{P}_{i0} = \boldsymbol{I}_{d_i}$, for $i \in [N]$ and $\boldsymbol{P}_{ij} = \boldsymbol{0}_p$, $\forall (i, j) \in [N] \times [N]$. With these choices, the Laplacian quadratic form of this sheaf is equal to the regularizer term

$$Q_{\mathcal{F}}(\boldsymbol{\theta}, \boldsymbol{\Pi}) = \sum_{i=1}^{N} \| \boldsymbol{\theta}_i - \boldsymbol{\Pi}_i \boldsymbol{\theta}_0 \|^2. \tag{23}$$

Replacing this into (9), we get the hybrid FL objective (Zhang et al., 2024, Eq. (6) given $\mu_1 = 0$).

## E  SCALABILITY OF SHEAF-FMTL

Complex datasets necessitate models with substantial parameter counts to achieve optimal performance. The sheaf-based approach scales to these high-dimensional settings by leveraging the projection mechanism to distill essential shared information, thereby reducing the effective dimensionality required for inter-client communication. This ensures that even as the underlying data complexity increases, the communication overhead remains manageable compared to the baseline dFedU. While `Sheaf-FMTL` offers significant advantages in terms of communication efficiency, it introduces additional computational and storage burdens due to the maintenance and updating of restriction maps as illustrated in Table 1. To ensure that `Sheaf-FMTL` remains scalable and efficient in handling complex and large-scale FL scenarios, the following strategies can be employed

- **Sparse restriction maps.** Sparsity constraints on the restriction maps can be implemented to reduce the number of active parameters, thereby lowering both storage and computational requirements.
- **Low-rank approximations.** Low-rank matrix approximations for restriction maps can be used to significantly decrease the computational complexity and storage footprint without substantially compromising performance.

## F  REAL-WORLD APPLICATIONS OF SHEAF-FMTL

In this appendix, we explore practical scenarios where task similarities are inherently defined within vector spaces, making them well-suited for the application of `Sheaf-FMTL`. By examining representation learning and feature vector similarities in multi-modal tasks, we illustrate how our sheaf-theoretic framework effectively captures and leverages complex task relationships. These examples showcase the applicability of `Sheaf-FMTL` in diverse FL environments.

### F.1  REPRESENTATION LEARNING AND EMBEDDING SPACES

In many ML applications, tasks are associated with high-dimensional data that can be effectively represented through embeddings in vector spaces. These embeddings capture semantic, syntactic, or feature-based relationships between tasks, facilitating the modeling of task similarities as vector operations. For example, if we consider the Natural Language Processing (NLP) field, then tasks such as sentiment analysis, topic classification, and named entity recognition can be embedded in a semantic space using techniques like Word2Vec or BERT. The proximity of these task vectors in

the embedding space reflects their semantic relatedness. Similar NLP tasks often share underlying linguistic structures. By representing these tasks as vectors, `Sheaf-FMTL` can capture and leverage their shared characteristics to enhance collaborative learning.

## F.2 FEATURE VECTOR SIMILARITIES IN MULTI-MODAL TASKS

In multi-modal learning scenarios, tasks often involve integrating and processing data from different modalities (e.g., text, image, audio). Task similarities can be defined based on the feature vectors extracted from these modalities, enabling `Sheaf-FMTL` to model interactions across diverse data sources. For example, if we consider multi-modal sentiment analysis, then tasks that analyze sentiment from text, images, and audio can have their respective feature vectors embedded in a unified vector space. The similarities between these feature vectors can indicate shared sentiment characteristics across modalities. `Sheaf-FMTL` can utilize these vector similarities to facilitate collaborative learning, enhancing sentiment detection accuracy by leveraging cross-modal information. Another example is healthcare applications, where tasks involving the analysis of patient data from various sources (e.g., medical imaging, electronic health records, genomic data) can define task similarities based on the integrated feature vectors representing different data modalities. By modeling these similarities in vector space, `Sheaf-FMTL` can improve personalized treatment recommendations through effective knowledge sharing across related healthcare tasks.

## G SUPPORTING LEMMAS

**Lemma G.1.**

$$\boldsymbol{\theta}^T L_{\mathcal{F}} \boldsymbol{\theta} = \sum_{e=(i,j)\in\mathcal{E}} \left\| \mathcal{F}_{i\trianglelefteq e}(\boldsymbol{\theta}_i) - \mathcal{F}_{j\trianglelefteq e}(\boldsymbol{\theta}_j) \right\|^2 = \sum_{e=(i,j)\in\mathcal{E}} \left\| \boldsymbol{P}_{ij}\boldsymbol{\theta}_i - \boldsymbol{P}_{ji}\boldsymbol{\theta}_j \right\|^2. \tag{24}$$

*Proof.* Using the block matrix structure of $L_{\mathcal{F}}$ from equation (3), we can expand the quadratic form as follows

$$
\begin{aligned}
\boldsymbol{\theta}^T L_{\mathcal{F}} \boldsymbol{\theta} \\
&= \sum_{i\in\mathcal{V}}\sum_{j\in\mathcal{V}} \boldsymbol{\theta}_i^T L_{i,j} \boldsymbol{\theta}_j \\
&= \sum_{i\in\mathcal{V}} \boldsymbol{\theta}_i^T \left( \sum_{j\in\mathcal{N}_i} \boldsymbol{P}_{ij}^T \boldsymbol{P}_{ij} \right) \boldsymbol{\theta}_i - 2 \sum_{e=(i,j)\in\mathcal{E}} \boldsymbol{\theta}_i^T \boldsymbol{P}_{ij}^T \boldsymbol{P}_{ji} \boldsymbol{\theta}_j \\
&= \sum_{e=(i,j)\in\mathcal{E}} \left( \boldsymbol{\theta}_i^T \boldsymbol{P}_{ij}^T \boldsymbol{P}_{ij} \boldsymbol{\theta}_i + \boldsymbol{\theta}_j^T \boldsymbol{P}_{ji}^T \boldsymbol{P}_{ji} \boldsymbol{\theta}_j - 2\boldsymbol{\theta}_i^T \boldsymbol{P}_{ij}^T \boldsymbol{P}_{ji} \boldsymbol{\theta}_j \right) \\
&= \sum_{e=(i,j)\in\mathcal{E}} \left( \left\| \boldsymbol{P}_{ij}\boldsymbol{\theta}_i \right\|^2 + \left\| \boldsymbol{P}_{ji}\boldsymbol{\theta}_j \right\|^2 - 2\left\langle \boldsymbol{P}_{ij}\boldsymbol{\theta}_i, \boldsymbol{P}_{ji}\boldsymbol{\theta}_j \right\rangle \right) \\
&= \sum_{e=(i,j)\in\mathcal{E}} \left\| \boldsymbol{P}_{ij}\boldsymbol{\theta}_i - \boldsymbol{P}_{ji}\boldsymbol{\theta}_j \right\|^2. \tag{25}
\end{aligned}
$$

Recalling that $\mathcal{F}_{i\trianglelefteq e}$ and $\boldsymbol{P}_{ij}$ are used interchangeably to denote the restriction map from vertex $i$ to edge $e = (i, j)$, we obtain

$$\boldsymbol{\theta}^T L_{\mathcal{F}} \boldsymbol{\theta} = \sum_{e=(i,j)\in\mathcal{E}} \left\| \boldsymbol{P}_{ij}\boldsymbol{\theta}_i - \boldsymbol{P}_{ji}\boldsymbol{\theta}_j \right\|^2 = \sum_{e=(i,j)\in\mathcal{E}} \left\| \mathcal{F}_{i\trianglelefteq e}(\boldsymbol{\theta}_i) - \mathcal{F}_{j\trianglelefteq e}(\boldsymbol{\theta}_j) \right\|^2. \tag{26}$$

$\square$

**Lemma G.2.**

$$\ker(L_{\mathcal{F}}) = \underset{\boldsymbol{\theta}\in C^0(\mathcal{F})}{\arg\min} \, Q_{\mathcal{F}}(\boldsymbol{\theta}) = \underset{\boldsymbol{\theta}\in C^0(\mathcal{F})}{\arg\min} \, \boldsymbol{\theta}^T L_{\mathcal{F}} \boldsymbol{\theta}. \tag{27}$$

*Proof.* Let $\boldsymbol{\theta} \in \ker(L_{\mathcal{F}})$. Then, $L_{\mathcal{F}}\boldsymbol{\theta} = 0$. By the definition of $Q_{\mathcal{F}}(\boldsymbol{\theta})$, we have

$$Q_{\mathcal{F}}(\boldsymbol{\theta}) = \boldsymbol{\theta}^T L_{\mathcal{F}} \boldsymbol{\theta} = 0.$$

Therefore, $\boldsymbol{\theta} \in \arg\min_{\boldsymbol{\theta} \in C^0(\mathcal{F})} Q_{\mathcal{F}}(\boldsymbol{\theta})$.

Conversely, let $\boldsymbol{\theta} \in \arg\min_{\boldsymbol{\theta} \in C^0(\mathcal{F})} Q_{\mathcal{F}}(\boldsymbol{\theta})$. Then, $Q_{\mathcal{F}}(\boldsymbol{\theta}) = 0$, which implies

$$0 = Q_{\mathcal{F}}(\boldsymbol{\theta}) = \sum_{e=(i,j)\in\mathcal{E}} \|\mathcal{F}_{i \trianglelefteq e}(\boldsymbol{\theta}_i) - \mathcal{F}_{j \trianglelefteq e}(\boldsymbol{\theta}_j)\|^2. \tag{28}$$

Thus, we get

$$\mathcal{F}_{i \trianglelefteq e}(\boldsymbol{\theta}_i) = \mathcal{F}_{j \trianglelefteq e}(\boldsymbol{\theta}_j) \quad \forall e = (i,j) \in \mathcal{E}. \tag{29}$$

This concludes the proof. $\qquad\square$

## H  PROOF OF THEOREM 3.5

We start by introducing the matrices $\boldsymbol{J}_{ij} \in \mathbb{R}^{d_{ij} \times d_i}$ having all its entries equal to one. Then, let us define the block matrix $\boldsymbol{H}$ such that $\boldsymbol{H}_{ij} = \boldsymbol{J}_{ij}$ if $(i,j) \in \mathcal{E}$, and $\boldsymbol{H}_{ij} = \boldsymbol{0}$, otherwise. Then, $\boldsymbol{\theta}$ and $\boldsymbol{P}$ can be updated using the following updates

$$\boldsymbol{\theta}^{k+1} = \boldsymbol{\theta}^k - \alpha(\nabla f(\boldsymbol{\theta}^k) + \lambda(\boldsymbol{P}^k)^T \boldsymbol{P}^k \boldsymbol{\theta}^k), \tag{30}$$

$$\boldsymbol{P}^{k+1} = \boldsymbol{H} \odot \left(\boldsymbol{P}^k - \eta\lambda \boldsymbol{P}^k \boldsymbol{\theta}^{k+1}(\boldsymbol{\theta}^{k+1})^T\right), \tag{31}$$

where $\odot$ is the Hadamard product and the matrix $\boldsymbol{H}$ is introduced to preserve the block structure of $\boldsymbol{P}$ by zeroing out the entries that do not correspond to edges in the graph.

Next, we analyze the convergence of the `Sheaf-FMTL` algorithm by studying the descent steps in $\boldsymbol{\theta}$ and $\boldsymbol{P}$ separately. This approach allows us to establish bounds on the decrease of the objective function $\Psi(\boldsymbol{\theta}, \boldsymbol{P})$ in each descent step.

**Theorem H.1.** *Let Assumptions 1 and 2 hold. Assume the learning rates $\alpha$ and $\eta$ satisfy the conditions $\alpha < \frac{2}{NL}$ and $\eta < \frac{2}{\lambda D_\theta^2}$, respectively. Then, the averaged gradient norm is upper bounded as follows*

$$\frac{1}{K}\sum_{k=0}^{K-1} \|\nabla\Psi(\boldsymbol{\theta}^k, \boldsymbol{P}^k)\|^2 \le \frac{1}{\rho K}(\Psi(\boldsymbol{\theta}^0, \boldsymbol{P}^0) - \Psi^\star), \tag{32}$$

*where $\rho = \min\left\{\alpha\left(1 - \frac{\alpha NL}{2}\right), \eta\left(1 - \frac{\eta\lambda D_\theta^2}{2}\right)\right\}$ and $\Psi^\star$ is the optimal value of $\Psi$.*

*Proof.* Using the Lipschitz continuity of the gradient of $f(\boldsymbol{\theta})$

$$f(\boldsymbol{\theta}^{k+1}) \le f(\boldsymbol{\theta}^k) + \langle\nabla f(\boldsymbol{\theta}^k), \boldsymbol{\theta}^{k+1} - \boldsymbol{\theta}^k\rangle + \frac{NL}{2}\|\boldsymbol{\theta}^{k+1} - \boldsymbol{\theta}^k\|^2. \tag{33}$$

Adding and subtracting $\frac{\lambda}{2}\boldsymbol{\theta}^{k+1}(\boldsymbol{P}^k)^T\boldsymbol{P}^k\boldsymbol{\theta}^{k+1}$ to both sides of the inequality, we get

$$\Psi(\boldsymbol{\theta}^{k+1}, \boldsymbol{P}^k)$$

$$= f(\boldsymbol{\theta}^{k+1}) + \frac{\lambda}{2}\boldsymbol{\theta}^{k+1}(\boldsymbol{P}^k)^T\boldsymbol{P}^k\boldsymbol{\theta}^{k+1}$$

$$\le f(\boldsymbol{\theta}^k) + \langle\nabla f(\boldsymbol{\theta}^k), \boldsymbol{\theta}^{k+1} - \boldsymbol{\theta}^k\rangle + \frac{NL}{2}\|\boldsymbol{\theta}^{k+1} - \boldsymbol{\theta}^k\|^2 + \frac{\lambda}{2}\boldsymbol{\theta}^{k+1}(\boldsymbol{P}^k)^T\boldsymbol{P}^k\boldsymbol{\theta}^{k+1}$$

$$= f(\boldsymbol{\theta}^k) + \langle\nabla f(\boldsymbol{\theta}^k), \boldsymbol{\theta}^{k+1} - \boldsymbol{\theta}^k\rangle + \frac{NL}{2}\|\boldsymbol{\theta}^{k+1} - \boldsymbol{\theta}^k\|^2 + \frac{\lambda}{2}\boldsymbol{\theta}^{k+1}(\boldsymbol{P}^k)^T\boldsymbol{P}^k\boldsymbol{\theta}^{k+1}$$

$$\quad - \frac{\lambda}{2}\boldsymbol{\theta}^k(\boldsymbol{P}^k)^T\boldsymbol{P}^k\boldsymbol{\theta}^k + \frac{\lambda}{2}\boldsymbol{\theta}^k(\boldsymbol{P}^k)^T\boldsymbol{P}^k\boldsymbol{\theta}^k$$

$$= f(\boldsymbol{\theta}^k) + \langle\nabla f(\boldsymbol{\theta}^k), \boldsymbol{\theta}^{k+1} - \boldsymbol{\theta}^k\rangle + \frac{NL}{2}\|\boldsymbol{\theta}^{k+1} - \boldsymbol{\theta}^k\|^2 + \frac{\lambda}{2}(\boldsymbol{\theta}^{k+1} - \boldsymbol{\theta}^k)^T(\boldsymbol{P}^k)^T\boldsymbol{P}^k\boldsymbol{\theta}^{k+1}$$

$$\quad + \frac{\lambda}{2}\boldsymbol{\theta}^k(\boldsymbol{P}^k)^T\boldsymbol{P}^k\boldsymbol{\theta}^k$$

$$= \Psi(\boldsymbol{\theta}^k, \boldsymbol{P}^k) + \langle\nabla_{\boldsymbol{\theta}}\Psi(\boldsymbol{\theta}^k, \boldsymbol{P}^k), \boldsymbol{\theta}^{k+1} - \boldsymbol{\theta}^k\rangle + \frac{NL}{2}\|\boldsymbol{\theta}^{k+1} - \boldsymbol{\theta}^k\|^2, \tag{34}$$

where the last equality follows from the definition of $\Psi(\boldsymbol{\theta}, \boldsymbol{P})$.

Using the update rule for $\boldsymbol{\theta}^{k+1}$, we have

$$
\begin{aligned}
&\langle \nabla_{\boldsymbol{\theta}} \Psi(\boldsymbol{\theta}^k, \boldsymbol{P}^k), \boldsymbol{\theta}^{k+1} - \boldsymbol{\theta}^k \rangle \\
&= \langle \nabla f(\boldsymbol{\theta}^k) + \lambda (\boldsymbol{P}^k)^T \boldsymbol{P}^k \boldsymbol{\theta}^k, -\alpha (\nabla f(\boldsymbol{\theta}^k) + \lambda (\boldsymbol{P}^k)^T \boldsymbol{P}^k \boldsymbol{\theta}^k) \rangle \\
&= -\alpha \|\nabla f(\boldsymbol{\theta}^k) + \lambda (\boldsymbol{P}^k)^T \boldsymbol{P}^k \boldsymbol{\theta}^k\|^2 \\
&= -\alpha \|\nabla_{\boldsymbol{\theta}} \Psi(\boldsymbol{\theta}^k, \boldsymbol{P}^k)\|^2.
\end{aligned}
\tag{35}
$$

On the other hand, we have

$$
\begin{aligned}
&\|\boldsymbol{\theta}^{k+1} - \boldsymbol{\theta}^k\|_2^2 \\
&= \alpha^2 \|\nabla f(\boldsymbol{\theta}^k) + \lambda (\boldsymbol{P}^k)^T \boldsymbol{P}^k \boldsymbol{\theta}^k\|^2 \\
&= \alpha^2 \|\nabla_{\boldsymbol{\theta}} \Psi(\boldsymbol{\theta}^k, \boldsymbol{P}^k)\|^2.
\end{aligned}
\tag{36}
$$

Substituting (35) and (36) back into (34), we obtain

$$
\Psi(\boldsymbol{\theta}^{k+1}, \boldsymbol{P}^k) \leq \Psi(\boldsymbol{\theta}^k, \boldsymbol{P}^k) - \alpha \left(1 - \frac{\alpha N L}{2}\right) \|\nabla_{\boldsymbol{\theta}} \Psi(\boldsymbol{\theta}^k, \boldsymbol{P}^k)\|^2.
\tag{37}
$$

By choosing $\alpha < \frac{2}{NL}$, we ensure that the term $1 - \frac{\alpha N L}{2}$ is positive.

From the definition of $\Psi(\boldsymbol{\theta}, \boldsymbol{P})$, we have

$$
\Psi(\boldsymbol{\theta}^{k+1}, \boldsymbol{P}^{k+1}) = f(\boldsymbol{\theta}^{k+1}) + \frac{\lambda}{2} (\boldsymbol{\theta}^{k+1})^T (\boldsymbol{P}^{k+1})^T \boldsymbol{P}^{k+1} \boldsymbol{\theta}^{k+1}.
\tag{38}
$$

Using the update rule for $\boldsymbol{P}^{k+1}$, i.e., $\boldsymbol{P}^{k+1} = \boldsymbol{H} \odot (\boldsymbol{P}^k - \eta \lambda \boldsymbol{P}^k \boldsymbol{\theta}^{k+1} (\boldsymbol{\theta}^{k+1})^T)$, we can write

$$
\begin{aligned}
&(\boldsymbol{P}^{k+1})^T \boldsymbol{P}^{k+1} \\
&= (\boldsymbol{H} \odot (\boldsymbol{P}^k - \eta \lambda \boldsymbol{P}^k \boldsymbol{\theta}^{k+1} (\boldsymbol{\theta}^{k+1})^T))^T (\boldsymbol{H} \odot (\boldsymbol{P}^k - \eta \lambda \boldsymbol{P}^k \boldsymbol{\theta}^{k+1} (\boldsymbol{\theta}^{k+1})^T)) \\
&\preceq (\boldsymbol{P}^k - \eta \lambda \boldsymbol{P}^k \boldsymbol{\theta}^{k+1} (\boldsymbol{\theta}^{k+1})^T)^T (\boldsymbol{P}^k - \eta \lambda \boldsymbol{P}^k \boldsymbol{\theta}^{k+1} (\boldsymbol{\theta}^{k+1})^T),
\end{aligned}
\tag{39}
$$

where the inequality follows from the fact that the Hadamard product with $\boldsymbol{H}$ zeros out some entries, which can only decrease the Frobenius norm.

Expanding the right-hand side, we get

$$
\begin{aligned}
&(\boldsymbol{P}^k - \eta \lambda \boldsymbol{P}^k \boldsymbol{\theta}^{k+1} (\boldsymbol{\theta}^{k+1})^T)^T (\boldsymbol{P}^k - \eta \lambda \boldsymbol{P}^k \boldsymbol{\theta}^{k+1} (\boldsymbol{\theta}^{k+1})^T) \\
&= (\boldsymbol{P}^k)^T \boldsymbol{P}^k - 2\eta\lambda (\boldsymbol{P}^k)^T \boldsymbol{P}^k \boldsymbol{\theta}^{k+1} (\boldsymbol{\theta}^{k+1})^T + \eta^2 \lambda^2 (\boldsymbol{P}^k)^T \boldsymbol{P}^k \boldsymbol{\theta}^{k+1} (\boldsymbol{\theta}^{k+1})^T \boldsymbol{\theta}^{k+1} (\boldsymbol{\theta}^{k+1})^T
\end{aligned}
\tag{40}
$$

Substituting this back into the expression for $\Psi(\boldsymbol{\theta}^{k+1}, \boldsymbol{P}^{k+1})$, we obtain

$$
\begin{aligned}
&\Psi(\boldsymbol{\theta}^{k+1}, \boldsymbol{P}^{k+1}) \\
&\leq f(\boldsymbol{\theta}^{k+1}) + \frac{\lambda}{2} (\boldsymbol{\theta}^{k+1})^T (\boldsymbol{P}^k)^T \boldsymbol{P}^k \boldsymbol{\theta}^{k+1} - \eta \lambda^2 (\boldsymbol{\theta}^{k+1})^T (\boldsymbol{P}^k)^T \boldsymbol{P}^k \boldsymbol{\theta}^{k+1} (\boldsymbol{\theta}^{k+1})^T \boldsymbol{\theta}^{k+1} \\
&\quad + \frac{\eta^2 \lambda^3}{2} (\boldsymbol{\theta}^{k+1})^T (\boldsymbol{P}^k)^T \boldsymbol{P}^k \boldsymbol{\theta}^{k+1} (\boldsymbol{\theta}^{k+1})^T (\boldsymbol{\theta}^{k+1}) (\boldsymbol{\theta}^{k+1})^T \boldsymbol{\theta}^{k+1}.
\end{aligned}
\tag{41}
$$

Since the gradient of $\Psi$ with respect to $\boldsymbol{P}$ is given by $\nabla_P \Psi(\boldsymbol{\theta}, \boldsymbol{P}) = \lambda \boldsymbol{P} \boldsymbol{\theta} \boldsymbol{\theta}^T$, we can compute the following norm

$$
\begin{aligned}
&\|\nabla_P \Psi(\boldsymbol{\theta}^{k+1}, \boldsymbol{P}^k)\|_F^2 \\
&= \mathrm{Tr}((\lambda \boldsymbol{P}^k \boldsymbol{\theta}^{k+1} (\boldsymbol{\theta}^{k+1})^T)^T (\lambda \boldsymbol{P}^k \boldsymbol{\theta}^{k+1} (\boldsymbol{\theta}^{k+1})^T)) \\
&= \lambda^2 \mathrm{Tr}(\boldsymbol{\theta}^{k+1} (\boldsymbol{\theta}^{k+1})^T (\boldsymbol{P}^k)^T \boldsymbol{P}^k \boldsymbol{\theta}^{k+1} (\boldsymbol{\theta}^{k+1})^T) \\
&= \lambda^2 (\boldsymbol{\theta}^{k+1})^T (\boldsymbol{P}^k)^T \boldsymbol{P}^k \boldsymbol{\theta}^{k+1} (\boldsymbol{\theta}^{k+1})^T \boldsymbol{\theta}^{k+1},
\end{aligned}
\tag{42}
$$

where we have used the cyclic nature of the trace operator $\mathrm{Tr}(\cdot)$.

Therefore, we have the following bound

$$\Psi(\boldsymbol{\theta}^{k+1}, \boldsymbol{P}^{k+1})$$
$$\leq \Psi(\boldsymbol{\theta}^{k+1}, \boldsymbol{P}^k) - \eta \left(1 - \frac{\eta\lambda}{2}\|\boldsymbol{\theta}^{k+1}\|^2\right) \|\nabla_P \Psi(\boldsymbol{\theta}^{k+1}, \boldsymbol{P}^k)\|^2$$
$$\leq \Psi(\boldsymbol{\theta}^{k+1}, \boldsymbol{P}^k) - \eta \left(1 - \frac{\eta\lambda D_\theta^2}{2}\right) \|\nabla_P \Psi(\boldsymbol{\theta}^{k+1}, \boldsymbol{P}^k)\|^2, \tag{43}$$

where we have used Assumption 2.

Hence, to ensure $\left(1 - \frac{\eta\lambda D_\theta^2}{2}\right)$ is positive, we choose the value of $\eta$ to be $\eta < \frac{2}{\lambda D_\theta^2}$. Combining the inequalities (37) and (43), we get

$$\Psi(\boldsymbol{\theta}^{k+1}, \boldsymbol{P}^{k+1})$$
$$\leq \Psi(\boldsymbol{\theta}^k, \boldsymbol{P}^k) - \alpha \left(1 - \frac{\alpha NL}{2}\right) \|\nabla_\theta \Psi(\boldsymbol{\theta}^k, \boldsymbol{P}^k)\|^2 - \eta \left(1 - \frac{\eta\lambda D_\theta^2}{2}\right) \|\nabla_P \Psi(\boldsymbol{\theta}^{k+1}, \boldsymbol{P}^k)\|^2. \tag{44}$$

Summing up these inequalities from $k = 0$ to $K - 1$, we obtain

$$\Psi(\boldsymbol{\theta}^K, \boldsymbol{P}^K)$$
$$\leq \Psi(\boldsymbol{\theta}^0, \boldsymbol{P}^0) - \alpha \left(1 - \frac{\alpha NL}{2}\right) \sum_{k=0}^{K-1} \|\nabla_\theta \Psi(\boldsymbol{\theta}^k, \boldsymbol{P}^k)\|^2 - \eta \left(1 - \frac{\eta\lambda D_\theta^2}{2}\right) \sum_{k=0}^{K-1} \|\nabla_P \Psi(\boldsymbol{\theta}^{k+1}, \boldsymbol{P}^k)\|^2. \tag{45}$$

Let $\Psi^\star$ be the optimal value of $\Psi$, and we define $\rho = \min\left\{\alpha\left(1 - \frac{\alpha NL}{2}\right), \eta\left(1 - \frac{\eta\lambda D_\theta^2}{2}\right)\right\}$. Then, rearranging the terms, we can write

$$\frac{1}{K} \sum_{k=0}^{K-1} \|\nabla \Psi(\boldsymbol{\theta}^k, \boldsymbol{P}^k)\|^2 \leq \frac{1}{\rho K}(\Psi(\boldsymbol{\theta}^0, \boldsymbol{P}^0) - \Psi(\boldsymbol{\theta}^K, \boldsymbol{P}^K)) \leq \frac{1}{\rho K}(\Psi(\boldsymbol{\theta}^0, \boldsymbol{P}^0) - \Psi^\star), \tag{46}$$

where we have used that $\Psi^\star \leq \Psi(\boldsymbol{\theta}^K, \boldsymbol{P}^K)$ and $\|\nabla \Psi(\boldsymbol{\theta}^k, \boldsymbol{P}^k)\|^2 = \|\nabla_{\boldsymbol{\theta}} \Psi(\boldsymbol{\theta}^k, \boldsymbol{P}^k)\|^2 + \|\nabla_{\boldsymbol{P}} \Psi(\boldsymbol{\theta}^k, \boldsymbol{P}^k)\|^2$. □

# I ADDITIONAL EXPERIMENTAL DETAILS

## I.1 DATASETS

A summary of the datasets and the tasks used in Section 4.2 is presented in Table 3. These datasets are real-world datasets created in federated environments with varying degrees of heterogeneity. A detailed description of the datasets along with their specific data partitioning schemes is provided in Table 4. To further quantify the Non-IIDness in our data partitions, we have incorporated quantitative metrics assessing the degree of Non-IIDness across different datasets in Table 5.

- **Rotated MNIST (R-MNIST).** Following similar techniques as outlined in (Liu et al., 2022a), we shuffle and then evenly separate the original MNIST dataset between 40 clients. Next, we randomly divide the clients into four groups, each containing 10 clients. We then apply rotations of $\{0°, 90°, 180°, 270°\}$ to each group respectively. Therefore, clients within the same group share identical image rotations, resulting in the formation of four distinct clusters. The MNIST dataset is available under the CC BY-SA 3.0 license.

- **Heterogeneous CIFAR-10 (H-CIFAR-10).** The original CIFAR-10 dataset is split among 30 clients, and heterogeneity is introduced by assigning each client a random number of samples from 5 randomly selected classes out of the 10 available classes, following a similar approach as in (T Dinh et al., 2020; Liu et al., 2022a). The CIFAR-10 dataset is available under the MIT license.

- **Human Activity Recognition.** The dataset is composed of data gathered from the accelerometers and gyroscopes of smartphones used by 30 individuals, each performing one of six activities: walking, walking upstairs, walking downstairs, sitting, standing, or lying down. In this dataset, the data from each individual/client is treated as a unique task, with the primary objective being to differentiate between these activities. To identify each activity appropriately, feature vectors with 561 elements representing various time and frequency domain variables are used in the analysis. The dataset (Anguita et al., 2013) is licensed under a Creative Commons Attribution 4.0 International (CC-BY 4.0) license.

- **Vehicle Sensor.** The dataset involves collecting data from a network of 23 wireless sensors, including acoustic (microphones), seismic (geophones), and infrared (polarized IR sensors), strategically placed along a specific road segment. This dataset aims to facilitate binary classification for identifying two types of vehicles: the Assault Amphibian Vehicle (AAV) and the Dragon Wagon (DW). Each sensor, treated as a unique task or client, gathers acoustic and seismic data encapsulated in a 100-dimensional feature vector, representing the recordings as vehicles pass by. The Vehicle dataset was originally made public by its authors as a research dataset (Duarte & Hu, 2004).

- **Google Glass Eating and Motion (GLEAM).** The dataset is collected using Google Glass from 38 individuals. It captures high-resolution sensor data to identify specific activities such as eating. This extensive dataset, consisting of 27,800 entries, each with a 180-dimensional feature vector, records head movements for binary classification to determine if the wearer is eating or not. The data includes accelerometer, gyroscope, and magnetometer readings, analyzed for statistical, spectral, and temporal characteristics to distinguish eating from other activities like walking, talking, and drinking. The GLEAM dataset, released by its original authors (Rahman et al., 2015), is available for non-commercial use.

- **School.** The dataset, originally introduced in (Goldstein, 1991), seeks to forecast the exam results of 15,362 students from 139 secondary schools. The dataset contains information for each school, with student numbers ranging from 22 to 251, and each student is described using a 28-dimensional feature vector. This vector contains information about the school's ranking, the student's birth year, and the availability of free meals at the school. The dataset has been made publicly available in (Zhou et al., 2011).

Table 3: Summary of the datasets and tasks used in our empirical setup.

| Dataset | Task | # Clients/Tasks | Input Dimension |
| --- | --- | --- | --- |
| R-MNIST | Classification | 40 | $28 \times 28 \times 1$ |
| H-CIFAR-10 | Classification | 30 | $32 \times 32 \times 3$ |
| HAR | Classification | 30 | 561 |
| Vehicle Sensor | Classification | 23 | 100 |
| GLEAM | Classification | 38 | 180 |
| School | Regression | 139 | 28 |

## I.2 EXPERIMENTAL SETTINGS

In the first experiment, we use a train/test split ratio of $75\%/25\%$ for all datasets as done in (Smith et al., 2017). For the classification task, the model used in all experiments is the multinomial logistic regression model with $L_2$-regularized cross-entropy as the loss function. Similar to (Dinh et al., 2022), we reduce the data size by $80\%$ for half of the clients to mimic the real-world FL setting where some clients have small datasets and can benefit from collaboration. For the regression task, we consider a linear model and the loss function to be the regularized $L_2$ loss. For dFedU, the weights $\{a_{ij}\}$, defined in (16), are taken to be $a_{ij} = 1$, $\forall (i,j) \in \mathcal{E}$.

For each experiment, we report both the average test accuracy/MSE and its corresponding one standard error shaded area based on five runs. Since all models have the same size $d$, we choose the projection space dimension to be $\gamma d$, where $\gamma \in (0, 1]$. In our experiments, the graph topology is based on the Erdős-Rényi model, where we randomly generate a network consisting of $N$ clients with a connectivity ratio $p = 0.15$ for Rotated MNIST and Heterogenous CIFAR-10 datasets, and

Table 4: Data partitioning strategies across datasets where $n_k$ is the number of training samples of client $k$.

| Dataset | Data split | Domain distribution | Label distribution |
|---|---|---|---|
| R-MNIST | min $n_k = 1500$ 
 max $n_k = 1500$ | 4 groups with distinct rotation angles | Uniform distribution within each rotation group |
| H-CIFAR-10 | min $n_k = 1515$ 
 max $n_k = 1839$ | Not explicitly divided into domains | Each client has data from 5 random classes |
| HAR | min $n_k = 210$ 
 max $n_k = 306$ | All activity classes per client | Uniform across activity classes within each client |
| Vehicle Sensor | min $n_k = 872$ 
 max $n_k = 1933$ | Same label set with different feature distributions | Balanced across vehicle classes per sensor |
| GLEAM | min $n_k = 699$ 
 max $n_k = 776$ | All activity classes with uniform distribution | Balanced between eating and non-eating classes |
| School | min $n_k = 15$ 
 max $n_k = 175$ | Shared regression task with uniform feature sets | Continuous targets with varying distributions per school |

Table 5: Degree of non-IIDness across datasets.

| Dataset | Non-IID Metric | Description |
|---|---|---|
| R-MNIST | Rotation angle variance | High domain heterogeneity with 4 distinct rotation groups |
| H-CIFAR-10 | Label distribution | High label distribution skew among clients |
| HAR | Inter-client variability | High heterogeneity due to unique individual data per client |
| Vehicle Sensor | Feature distribution | High feature heterogeneity across different sensors |
| GLEAM | Label distribution | Low heterogeneity with balanced labels |
| School | Continuous targets variance | Moderate to high heterogeneity based on inter-school performance variability |

$p = 0.2$ for the rest of datasets. Both learning rates are chosen small to satisfy the conditions in Theorem 3.5. Furthermore, `Sheaf-FMTL` uses the same global learning rate ($\alpha$) as dFedU to update the models for a fair comparison. The linear maps are initialized randomly from a normal distribution with a mean of 0 and a variance of 1. The values of the regularization parameter ($\lambda$) used for every dataset are listed in Table 6.

For the second experiment, we generate the modified datasets as follows. To mimic the fact that each client has a different model size, we randomly drop a set of features with a drop factor sampled from the uniform distribution $\mathcal{U}([0, 0.4])$. As an illustration, the model size distribution across the number of clients in the modified Vehicle dataset is plotted in Figure 4.

Table 6: Regularization parameters used during training.

| Dataset | HAR | Vehicle | GLEAM | School | R-MNIST | H-CIFAR-10 |
|---|---|---|---|---|---|---|
| Regularization parameter ($\lambda$) | 0.05 | 0.001 | 0.001 | 0.01 | 0.001 | 0.001 |

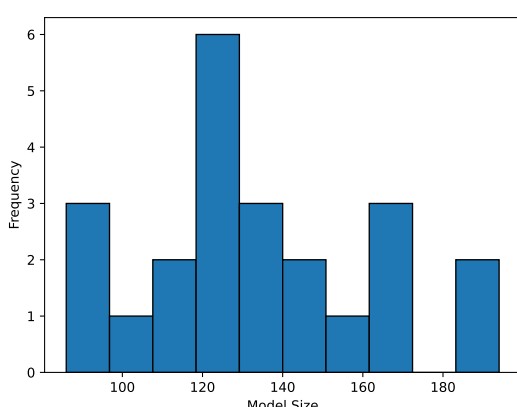

Figure 4: Model size distribution across clients in the modified Vehicle dataset.

### I.3 HARDWARE & CODE

Our experiments were carried out on a system equipped with an Intel(R) Xeon(R) CPU operating at 2.20GHz with 2 cores and 12 GB of RAM. The algorithms are implemented in Python using PyTorch (Paszke et al., 2019), and NetworkX (Hagberg et al., 2008).

## J  SUPPLEMENTARY EXPERIMENTAL RESULTS

Figure 5 compares the performance of our proposed `Sheaf-FMTL` method with dFedU on four datasets: (a) the Vehicle dataset, (b) the School dataset, (c) the HAR dataset and (d) the GLEAM dataset using $\gamma = \{0.1, 0.3\}$. The experiments evaluate the performance of the model in terms of communication rounds and transmitted bits. For instance, `Sheaf-FMTL` requires more communication rounds for the Vehicle dataset to achieve similar test accuracy compared to dFedU. As the number of communication rounds increases, the performance gap between the two methods narrows. However, when examining the number of transmitted bits, `Sheaf-FMTL` demonstrates a clear advantage, requiring fewer bits to reach higher accuracy levels. For example, `Sheaf-FMTL` with $\gamma = 0.1$ reaches a test accuracy of 0.85 with just 72 transmitted Kbits, while dFedU requires over 450 Kbits to approach this accuracy. Similar trends are observed for the other datasets. `Sheaf-FMTL` recovers the same performance as dFedU in terms of test accuracy with a slight drop in test accuracy for the HAR and School datasets. Hence, the sheaf-based approach effectively captures the heterogeneous relationships among clients, achieving the same test accuracy using fewer transmitted bits than the baseline dFedU.

## K  LIMITATIONS AND FUTURE WORK

In this section, we highlight some of the limitations of our approach, while outlining several potential directions for future work.

**Dimension of the projection space.** The current work assumes a fixed dimension for the projection spaces associated with the edges in the sheaf structure. However, the optimal dimension may depend on factors such as the complexity of the client models and the relationships between the tasks. Future work could explore methods for automatically learning or finding the dimension of the projection spaces.

**Restriction maps.** The proposed framework uses linear maps for the restriction and lifting operations between the client and edge spaces. While this allows for efficient computation, it may limit the ability to capture complex relationships between the tasks. Investigating the use of non-linear restriction maps could potentially improve the capabilities of the sheaf-based FMTL framework.

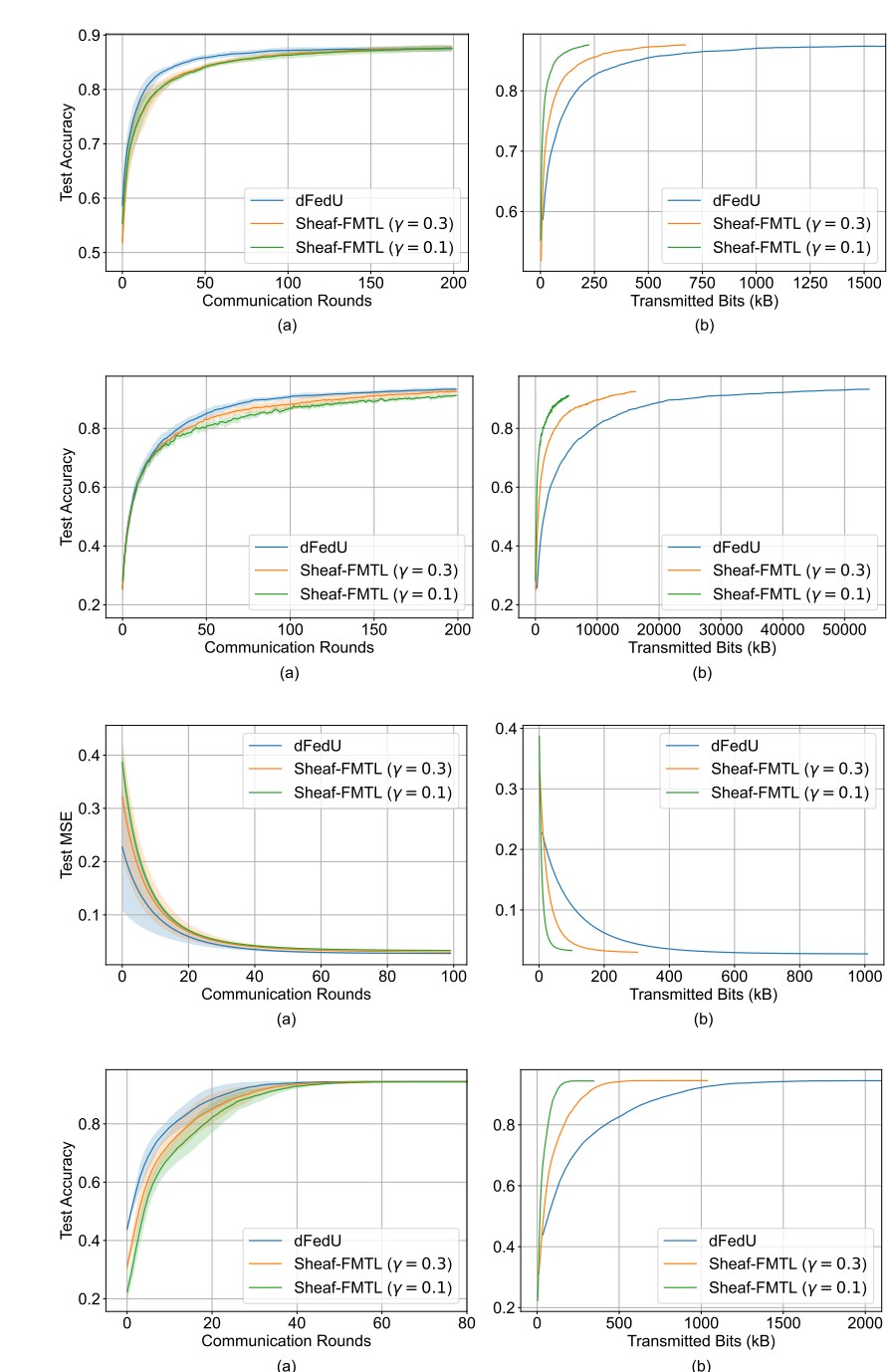

Figure 5: Test/MSE accuracy as a function of the number of communication rounds and the number of transmitted bits for the Vehicle dataset (first row), the School dataset (second row), the HAR dataset (third row), and the GLEAM dataset (bottom).

**Storage requirement of the restriction maps.** Our proposed method requires the storage of additional $\boldsymbol{P}_{ij}$ matrices that are of size $d_i \times d_{ij}$. While this may not be considered an issue in a cross-silo FL setting where the clients or organizations involved typically have more substantial computational and storage capabilities, this can be seen as a limitation to clients in massively distributed FL scenarios, e.g., mobile devices or IoT devices.

**Relaxing Assumption 2.** The theoretical analysis in the paper relies on Assumption 2, which assumes the models are bounded. Relaxing or removing this assumption would broaden the applicability of the theoretical results.

