# OpenReview forum: "Tackling Feature and Sample Heterogeneity in Decentralized Multi-Task Learning: A Sheaf-Theoretic Approach"
_ICLR.cc/2025/Conference — Submitted to ICLR 2025_

### Official Review · Reviewer_RmE8 · 2024-10-30

**Soundness:** 2
**Presentation:** 1
**Contribution:** 2
**Rating:** 5
**Confidence:** 2

**Summary:**

Federated multi-task learning is an important framework that deals with multiple tasks simultaneously in federated learning. The authors claim that existing methods fail to capture complex relationships. This paper proposes a so-called "sheaf-theoretic approach".

**Strengths:**

This method is indeed an interesting work, which provides a different perspective.

**Weaknesses:**

1. I do not understand the limitations of existing FMTL methods. What do you mean by "without a deeper complex interdependencies between tasks"? What is the impact of such deficiencies?

2. The claimed convergence rate is $O(1/K)$, which lacks comparison with existing methods. Hope that authors can provide detailed comparison with the state-of-the-art FMTL methods.

3. For numerical experiments, more comparison with other FMTL methods are needed.

4. In general, I feel that this paper is hard to follow. I do not understand the sheaf-theoretic approach.

**Questions:**

How to get eq.(8)? I do not follow. Hope that the authors can provide intuition and step-to-step derivation of eq.(8).

---

> ### Author Response · Authors · 2024-11-20
> **Response to Reviewer RmE8 (Part 1)**
>
> > I do not understand the limitations of existing FMTL methods. What do you mean by "without a deeper complex interdependencies between tasks"? What is the impact of such deficiencies?
>
> **Response:**
>
> We apologize for the confusion caused by our phrasing regarding the limitations of existing FMTL methods. Let us clarify the intended meaning and elaborate on the impact of these limitations.
> - **Clarification of "Deeper Complex Interdependencies":**
> In the context of FMTL, interdependencies between tasks refer to the relationships and interactions that exist among the different tasks being learned across clients. Traditional FMTL frameworks often model these interdependencies using simple and fixed scalar weights that quantify the pairwise relationships between tasks. While this approach captures the basic notion of task relatedness, it fails to represent more intricate and higher-order relationships that may exist in real-world scenarios. On the other hand, our sheaf-theoretic-based approach addresses these limitations by modeling task interdependencies through cellular sheaves, which inherently capture higher-order and more complex relationships between tasks. This allows for a more nuanced and flexible representation, enabling the model to adapt dynamically to the intricate patterns of interdependence that exist in decentralized and heterogeneous environments. For example, consider a scenario with multiple clients working on related but distinct tasks, such as image classification for different object categories. Symmetric scalar weights may not adequately capture the relationships between these tasks, as certain tasks might share more nuanced feature representations or hierarchical dependencies that scalar weights cannot encapsulate.
> - **Impact of Such Deficiencies:**
>    - **Limited representation of task relationships:** By using fixed scalar weights, existing methods oversimplify the complex relationships between tasks, potentially overlooking subtle but significant interdependencies. Furthermore, fixed weights do not adapt dynamically to evolving task relationships during the training process, which can hinder the model's ability to leverage emerging patterns of interdependence.
>    - **Suboptimal model performance:** Inadequate modeling of task interdependencies can lead to inefficient knowledge transfer between clients, resulting in poorer generalization performance on individual tasks.
>
> **To address your concerns, we added more detail in the introduction to alleviate potential confusion. We also invite the reviewer to visit Appendix F** where we discuss scenarios where task similarities are inherently defined in vector spaces, such as through representation learning and multi-modal feature integrations.
>
> > The claimed convergence rate is $\mathcal{O}(1/K)$ , which lacks comparison with existing methods. Hope that authors can provide detailed comparison with the state-of-the-art FMTL methods.
>
> **Response:**
>
> We thank the reviewer for the comment. Sheaf-FMTL achieves the same $\mathcal{O}(1/K)$ convergence rate as state-of-the-art FMTL methods (MOCHA and dFedU). Beyond matching existing convergence guarantees, our framework offers several notable advantages:
> - Sheaf-FMTL provides a unified framework for FMTL, as detailed in Appendix D, integrating both data and model heterogeneity seamlessly.
> - We go beyond modeling the similarity of the tasks using simple scalars, through the restriction maps, capturing more complex interactions.
> - By sharing only lower-dimensional projections of model parameters rather than full models, Sheaf-FMTL inherently enhances privacy protection, preventing the reconstruction of complete models from transmitted data. This projection also leads to reduced communication costs compared to methods like dFedU.
> - Unlike conventional FMTL algorithms, Sheaf-FMTL adeptly handles varying model sizes across clients, making it highly effective in federated environments where client datasets and model architectures differ significantly.
>
> **To take into consideration the reviewer's comment, we have also updated our list of contributions to emphasize that Sheaf-FMTL recovers the same convergence rate as the state-of-the-art FMTL methods, underlining its theoretical soundness alongside its practical benefits.**

---

> ### Author Response · Authors · 2024-11-20
> **Response to Reviewer RmE8 (Part 2)**
>
> > For numerical experiments, more comparison with other FMTL methods are needed.
>
> **Response:**
>
> We appreciate the reviewer's observation regarding the number of methods used for comparison in our study. To the best of our knowledge, dFedU is the state-of-the-art method in FMTL in the decentralized setting, and hence it serves as a strong baseline. Most of the other FMTL algorithms require the presence of a parameter server, which is why we restrict the comparison to dFedU. Given that both dFedU and Sheaf-FMTL operate in decentralized settings without relying on a central server, dFedU provides a pertinent point of comparison. Finally, we are willing to compare to other FMTL algorithms in the decentralized setting if the reviewer thinks that such algorithms exist.
>
> > In general, I feel that this paper is hard to follow. I do not understand the sheaf-theoretic approach.
>
> **Response:**
>
> We thank the reviewer for their feedback regarding the clarity of our manuscript, specifically concerning the sheaf-theoretic approach employed in our algorithm. We acknowledge that sheaf theory, being a specialized area within mathematics, can present challenges to readers unfamiliar with its concepts. In fact, the sheaf-theoretic approach introduces the concept of cellular sheaves, which assigns data to the vertices and edges of a graph, along with linear maps (restriction maps) that describe how data on vertices is related to data on edges. In FMTL, the vertices represent clients, and the edges represent the relationships between clients. The local models $\theta_i$ are assigned to the vertices, and the restriction maps $P_{ij}$ describe how the local models are related through the edges.
> Now, we provide more explanation by examining Figure 1 of the paper. Let's restrict the discussion to clients $i$ and $k$:
> - The stalks $\theta_i$ and $\theta_k$ represent the local models of clients $i$ and $k$, respectively.
> - The restriction maps $P_{ik}$ and $P_{ki}$ project the local models onto the edge space $\mathcal{F}(e_1)$, capturing the shared features or relationships between the clients.
> - The sheaf Laplacian term $\frac{\lambda}{2} \theta^T L_{\mathcal{F}}(P) \theta$ enforces consistency between the projections of the local models onto the edge space, promoting collaboration among the clients.
>
> **To this end, we added a paragraph after Lemma 3.2 in the revised manuscript to specify how our sheaf-theoretic approach is used in the context of FMTL. A more in-depth discussion on the sheaf-theoretic approach for FMTL can be found in Appendix A.**
>
> > How to get eq.(8)? I do not follow. Hope that the authors can provide intuition and step-to-step derivation of eq.(8).
>
> **Response:**
>
> We appreciate the reviewer's feedback and understand the importance of eq. (8) in our manuscript and appreciate the opportunity to clarify its derivation and intuitive significance. In fact, eq. (8) generalizes the objective considered in the dFedU paper (eq. (1)) by considering a sheaf Laplacian regularization term instead of the graph Laplacian regularizer considered in the dFedU paper. Moreover, many MTL problems can be captured via this formulation [Ref1-Ref4]. **To make it clear, we added a paragraph before eq. (8) to motivate and provide more intuition on how we got this equation.**
>
> ---
> [Ref1] L. Jacob, J.-p. Vert, and F. R. Bach. "Clustered multi-task learning: A convex formulation". In Neural Information Processing Systems, 2009.
>
> [Ref2] J. Zhou, J. Chen, and J. Ye. "Clustered multi-task learning via alternating structure optimization". In Neural Information Processing Systems, 2011.
>
> [Ref3] Smith, V., Chiang, C. K., Sanjabi, M., and Talwalkar, A. S.. "Federated multi-task learning", Advances in neural information processing systems, 30, 2017.
>
> [Ref4] Dinh, Canh T., Tung T. Vu, Nguyen H. Tran, Minh N. Dao, and Hongyu Zhang. "A new look and convergence rate of federated multitask learning with laplacian regularization." IEEE Transactions on Neural Networks and Learning Systems, 2022.

---

> ### Comment · Reviewer_RmE8 · 2024-11-21
> **Further discussion**
>
> Thanks for your response. For the convergence rate, if my understanding is correct, the previous O(1/K) convergence rate comes from Lemma 1 in Dinh et al. 2022, which bounds the sum of squared norm of gradients. The bound in Dinh et al. 2022 is $\sigma_2^2$. Therefore the averaged value of sum of squared norm of gradients is $\sigma_2^2/N$. ($N$ in Dinh et al.2022 is $K$ in your paper). I suggest to mention it in your paper so that readers can easily find the baseline.
>
> Moreover, a convergence rate that is same as existing work does not demonstrate the advantage of your proposed method. Since you argue that your method has better performance under data/model heterogeneity, or varying model sizes across clients, could you please establish some theoretical bounds that are better than Dinh et al. 2022 under some heterogeneity assumptions (you may make some new assumptions on the degree of heterogeneity)? This will significantly improve the soundness of this paper.

---

> ### Author Response · Authors · 2024-11-21
> **Response to further discussion**
>
> We thank the reviewer for the comment. First, we would like to clarify that Lemma 1 in (Dinh et al. 2022) establishes the relationship between the sum of the squared norms of the gradients and the sum of the squared norms of the gradients of the global FMTL objective function with respect to the local models. This lemma is leveraged to prove the convergence rate presented in Theorem 2. Importantly, our analysis **generalizes** the framework established by (Dinh et al. 2022) through the following key considerations:
> - **Heterogeneous model architectures.** While (Dinh et al. 2022) assumed **uniform model dimensions** across all clients, our sheaf-theoretic approach inherently supports **heterogeneous model architectures** and **varying feature spaces**. This generalization is significant because it addresses more realistic and complex federated environments where clients may possess models of differing sizes and feature spaces.
> - **Learning task similarities via optimization of restriction maps.**
>     A core innovation of our approach lies in **dynamically learning the task similarities between clients** by optimizing the restriction maps $P_{ij}$ unlike (Dinh et al. 2022) that assumes **known or uniform task relationships**. These restriction maps serve as projection matrices that map local models onto shared interaction spaces, effectively capturing the underlying relationships and dependencies between different client tasks. By jointly optimizing both the local models $\theta_i$ and the restriction maps $P_{ij}$, our framework allows for **adaptive relationship modelling**. In fact, the restriction maps are not fixed but are learned during the training process, enabling the model to adaptively discover and leverage the inherent similarities and complementarities between client tasks. This dynamic learning of task relationships facilitates more nuanced and effective collaboration among clients.
>
> Furthermore, as highlighted in our response, our approach offers additional advantages over (Dinh et al. 2022) in terms of **communication efficiency** and **enhanced privacy preservation**:
>
> - **Communication efficiency.** Our method reduces the communication overhead by transmitting significantly fewer bits compared to baseline methods. This is achieved through the compact representation of shared information via restriction maps, which are optimized to capture only the essential task similarities.
>
> - **Privacy preservation.** By exchanging projections of local models rather than the models themselves, our approach provides an additional layer of privacy. This ensures that sensitive raw data and model parameters remain more secure, addressing privacy concerns inherent in FL environments.

---

> ### Author Response · Authors · 2024-11-25
> **Follow-Up on Rebuttal**
>
> Dear Reviewer RmE8,
>
> One more time, we would like to express our gratitude for taking the time to review our paper. Your feedback has been extremely useful in refining and improving the work.
>
> We have carefully addressed all the comments and concerns raised in your initial review and we wanted to kindly ask if you could take another look at our responses to ensure that all your concerns have been adequately addressed. If there are any additional questions or points you'd like us to clarify, kindly let us know as the rebuttal phase is ending soon.
>
> If you find that the revisions have satisfactorily addressed your concerns, we would be grateful if you could consider updating your review score accordingly.

---

> ### Author Response · Authors · 2024-12-02
> **New baselines**
>
> First, we still believe that for a fair comparison, we need to compare our algorithm to a baseline that **leverages task diversity** and operates in **a decentralized manner without relying on a parameter server**. However, since most reviewers criticized the comparison to only dFedU, we have extended our experimental evaluation by incorporating three state-of-the-art personalized federated learning (PFL) algorithms: DITTO, pFedMe, and Per-FedAvg. These algorithms are well-regarded in the literature for addressing heterogeneity in FL environments. We conducted additional experiments on two datasets—Vehicle Sensor and HAR—to evaluate the performance of our method against these baselines. The test accuracy is averaged over 5 runs reporting both mean and standard deviation.
>
> The results in the table below demonstrate that our proposed algorithm consistently achieves the highest test accuracy, comparable to or exceeding that of dFedU, while significantly reducing the number of transmitted bits. The new baselines (DITTO, pFedMe, Per-FedAvg) achieve lower communication overheads due to their star topology, leveraging a central parameter server.
>
> | Dataset        |             Algorithm             | Test Accuracy | Transmitted Bits (kB) |
> |----------------|:---------------------------------:|:-------------:|:---------------------:|
> | Vehicle Sensor |    Sheaf-FMTL ($\gamma$ = 0.1)    |  87.6 ± 0.61  |          224          |
> |                |    Sheaf-FMTL ($\gamma$ = 0.3)    |  87.6 ± 0.52  |          672          |
> |                |               dFedU               |  87.52 ± 0.53 |         2262.4        |
> |                |               DITTO               |  84.17 ± 1.66 |         161.6         |
> |                |               pFedme              |  84.82 ± 0.65 |         161.6         |
> |                |             Per-FedAvg            |  81.85 ± 1.16 |         161.6         |
> | HAR            |    Sheaf-FMTL ($\gamma$ = 0.1)    |  92.49 ± 1.21 |          5392         |
> |                |    Sheaf-FMTL ($\gamma$ = 0.3)    |  92.82 ± 0.51 |         16176         |
> |                |               dFedU               |  93.29 ± 0.69 |         53952         |
> |                |               DITTO               |  91.86 ± 1.09 |         2697.6        |
> |                |               pFedme              |  91.63 ± 1.15 |         2697.6        |
> |                |             Per-FedAvg            |  89.73 ± 0.78 |         2697.6        |

---

### Official Review · Reviewer_NGQr · 2024-11-03

**Soundness:** 2
**Presentation:** 3
**Contribution:** 2
**Rating:** 3
**Confidence:** 4

**Summary:**

This paper studied federated multi-task learning (FMTL) and presented an approach to incorporate the principles from sheaf theory to understand the interdependencies between the tasks. Given an underlying topology of client relationships, the paper utilizes the notion of a cellular sheaf to model the interactions between clients in a decentralized FMTL setting. The modeling approach considers similarity through the use of a projection operator, which is integrated into the learning process as a regularization term. The paper then introduces an alternating gradient descent algorithm that addresses the unknowns (projection matrices and reward parameters) by iteratively fixing one variable and solving for the other.

**Strengths:**

The paper addresses federated multi-task learning, a growing area of interest due to its collaborative nature that enhances the effectiveness of the learning process.

**Weaknesses:**

Given that the proposed solution lacks theoretical guarantees on regret and that there are existing empirical approaches that effectively capture heterogeneous agents, its significance is somewhat limited.

The paper also fails to provide a detailed comparison with benchmark approaches, despite the fact that federated multi-task learning has been extensively studied in the literature.

The theoretical result on bounded gradient is not very significant under the assumptions on smoothness in the paper.

**Questions:**

1. An alternating gradient descent approach for a regularized cost function is a common technique in constrained learning. With this in mind, could you comment on the primary innovation and challenges addressed of the paper?

2. What are some scenarios where the task similarities are defined in terms of vector spaces as required in the paper? Often task similarities are modeled in the form communication network among the tasks, clustering, or specific structures on the features.

3. Can you provide practical examples and physical interpretations of how the sheaf-theoretic approach is applied to real-world problems? Specifically, could you share examples of graph models with fixed-space node features in practical scenarios?

4. Additionally, how is the sheaf-theoretic approach, specifically the interaction between the tasks, integrated into the experiments involving real-world data?

4. The significance of the proposed approach is unclear in the absence of a regret bound.

5. What is the rationale for selecting dFedU as the sole benchmark for comparison, especially considering that the area has been extensively explored?

6. If considered for larger rounds, will the performance of solving the problems individually have a similar MSE in Fig 3b? If so, given that this also saves the communication cost, what is the advantage of the proposed approach over the naive approach of solving the tasks separately?

**Details Of Ethics Concerns:**

None.

---

> ### Author Response · Authors · 2024-11-20
> **Response to Reviewer NGQr (Part 1)**
>
> > An alternating gradient descent approach for a regularized cost function is a common technique in constrained learning. With this in mind, could you comment on the primary innovation and challenges addressed of the paper?
>
> **Response:**
>
> Thank you for your comment. We appreciate the opportunity to clarify the primary innovations and the specific challenges addressed by our paper. As pointed out by the reviewer, the alternating gradient descent approach for a regularized cost function is a widely recognized technique in constrained learning. The main innovation of our paper goes beyond using the alternating gradient descent approach as a tool **to solve the unified problem formulation we are proposing for FMTL by incorporating concepts from sheaf theory**. In what follows, we highlight the primary innovations and challenges addressed by our paper:
> - **Learning task similarity via restriction maps.** Sheaf-FMTL introduces learned restriction maps that dynamically capture the relationships between different tasks and models. These maps are governed by sheaf Laplacian regularization, which enforces consistency and smoothness across the sheaf structure, thereby enabling efficient knowledge sharing.
>
>     **Challenge addressed.** In federated environments, especially those with high-dimensional models and complex task relationships, maintaining consistency and effective knowledge transfer is challenging. Traditional regularization techniques may not adequately capture these nuanced dependencies. The sheaf Laplacian regularization in Sheaf-FMTL provides a principled way to learn and enforce these relationships, ensuring that updates across clients are both meaningful and coherent.
> - **Unified FMTL framework addressing both data and model heterogeneity.**  Sheaf-FMTL offers a unified framework that simultaneously addresses data heterogeneity (non-IID data distributions across clients) and model heterogeneity (varying model architectures or parameter spaces). This is achieved through the flexible sheaf-based modeling of interactions, which can adapt to both types of heterogeneity seamlessly.
>
>    **Challenge addressed.** Most existing FMTL approaches tend to focus on either data heterogeneity or model heterogeneity but not both simultaneously. Handling both complexities is essential for real-world federated learning scenarios where clients not only possess diverse data but also operate with different model architectures. Sheaf-FMTL's ability to model and manage both dimensions within a single coherent framework represents a significant advancement over traditional methods.
>
> > What are some scenarios where the task similarities are defined in terms of vector spaces as required in the paper? Often task similarities are modeled in the form communication network among the tasks, clustering, or specific structures on the features.
>
> **Response:**
>
> Thank you for raising an important point regarding the modeling of task similarities within our Sheaf-FMTL framework. We acknowledge that task similarities are often represented through various structures such as communication networks, clustering, or feature-specific constructs. However, there exist several scenarios where task similarities can naturally be defined in terms of vector spaces. **In Appendix F, we mention a few of these scenarios and demonstrate how they align with the requirements of our framework**.

---

> ### Author Response · Authors · 2024-11-20
> **Response to Reviewer NGQr (Part 2)**
>
> > Can you provide practical examples and physical interpretations of how the sheaf-theoretic approach is applied to real-world problems? Specifically, could you share examples of graph models with fixed-space node features in practical scenarios?
>
> **Response:**
>
> Thank you for your valuable feedback and for highlighting the need to explain the practical applications and physical interpretations of the sheaf-theoretic approach within our Sheaf-FMTL framework. To this end, we present real-world examples from two domains in Appendix F. Furthermore, we present other practical scenarios where the sheaf-theoretic approach is effectively applied, particularly focusing on graph models with fixed-space node features as illustrated in [Ref0].
> 1) **Multi-Camera Computer Vision Systems:**
>     - **Scenario.** In a computer vision task, multiple cameras are deployed to monitor a dynamic scene, such as counting coins on a table. These cameras may vary in terms of lighting conditions, perspectives, and sensor modalities (e.g., RGB cameras and LIDAR sensors), leading to heterogeneous data outputs that need to be fused for accurate object counting.
>     - **Sheaf-Theoretic application.** Each camera is represented as a node with a fixed feature vector derived from its specific data output:
>          - Uniform Lighting and Perspective Cameras: Feature vectors correspond directly to image data ($\mathbb{R}^{3mn}$, where $m$ and $n$ denote image dimensions).
>          - Varying Lighting and Perspective Cameras: Feature vectors include adjusted image representations to account for differences ($\mathbb{R}^{3mn}$ with preprocessing).
>          - Heterogeneous Sensors (e.g., Camera and LIDAR): Feature vectors encapsulate distinct data types, such as visual images and spatial point clouds ($\mathbb{R}^{3mn} \times \mathbb{R}^p$).
> Edges capture the spatial overlaps and interactions between different camera views. A sheaf is constructed where each node's fixed feature vector is associated with its camera domain, and restriction maps ensure consistency in overlapping regions. This sheaf structure facilitates the fusion of heterogeneous data, enabling Sheaf-FMTL to accurately count and identify objects by integrating complementary information from multiple sensor modalities.
>
> **Physical interpretation.** Applying sheaf theory to multi-camera systems with fixed-space node features allows for the seamless integration of diverse and overlapping data streams. The sheaf structure ensures that inconsistencies due to varying lighting conditions or different sensor modalities are reconciled through restriction maps, enabling accurate and reliable object counting. This approach enhances the robustness of computer vision systems by leveraging the collective strengths of multiple sensors while maintaining data integrity across heterogeneous inputs.
>
> 2) **Distributed Sensor Systems in Smart Grids**
>     - **Scenario.** $n$ smart grid applications, various distributed sensors monitor electrical parameters such as voltage, current, and frequency across different nodes in the grid. These sensors provide real-time measurements essential for maintaining grid stability and detecting anomalies.
>     - **Sheaf-Theoretic application.** Each sensor is represented as a node with a fixed feature vector containing its specific electrical measurements:
>          - Voltage Sensors: Feature vectors include voltage levels at specific points in the grid ($\mathbb{R}^n$).
>          - Current Sensors: Feature vectors capture current flow through transmission lines ($\mathbb{R}^m$).
>          - Frequency Sensors: Feature vectors monitor frequency stability at various grid nodes ($\mathbb{R}^p$).
> Edges denote the physical transmission lines or distribution paths connecting different sensor nodes. A sheaf is constructed where each node's fixed feature vector corresponds to its sensor's measurement space, and restriction maps ensure consistency across interconnected sensors. This sheaf structure allows Sheaf-FMTL to integrate diverse electrical measurements coherently, facilitating accurate state estimation and anomaly detection within the smart grid.
>
> **Physical interpretation.** Applying sheaf theory to distributed sensor systems in smart grids enables the seamless integration of heterogeneous electrical measurements across the grid's topology. The sheaf structure ensures that voltage, current, and frequency measurements are consistent across interconnected sensors, enhancing the accuracy of state estimations and the detection of grid anomalies. This approach improves the reliability and efficiency of smart grids by leveraging comprehensive and consistent data fusion from diverse sensor types while maintaining data integrity across the complex electrical network.
>
> ---
> [Ref0] Robinson, M. "Sheaves are the canonical data structure for sensor integration". Information Fusion, 36, pp.208-224, 2017.

---

> ### Author Response · Authors · 2024-11-20
> **Response to Reviewer NGQr (Part 3)**
>
> > Additionally, how is the sheaf-theoretic approach, specifically the interaction between the tasks, integrated into the experiments involving real-world data?
>
> **Response:**
>
> We thank the reviewer for the question. To integrate the sheaf-theoretic approach into our experiments, we model the interactions between tasks using cellular sheaves. Specifically, we represent each client as a node in a graph, with edges representing the interactions or relationships between clients. The feature space at each node includes the local data and model parameters of the client. The interaction space captures the shared or comparable features between clients, facilitating the comparison and integration of local models. The restriction maps project the local models onto the interaction space, enabling meaningful comparisons and collaborations between clients. These maps act as feature selection mechanisms, identifying common or comparable features between clients. In our experiments, we design the restriction maps to capture the shared information between clients, such as common features in the Rotated MNIST and Heterogeneous CIFAR-10 datasets, or shared patterns in the Vehicle and School datasets. We incorporate the sheaf Laplacian regularization term into our optimization objective, which enforces consistency between the projections of the local models onto the interaction space. This regularization term promotes collaboration between clients, encouraging them to leverage shared information to improve their local models. The sheaf Laplacian quadratic form measures the total discrepancy between the projections of the local models, guiding the optimization process to minimize this discrepancy.
>
> > What is the rationale for selecting dFedU as the sole benchmark for comparison, especially considering that the area has been extensively explored?
>
> **Response:**
>
> We appreciate the reviewer's observation regarding the number of methods used for comparison in our study. To the best of our knowledge, dFedU is the state-of-the-art method in FMTL in the decentralized setting, and hence it serves as a strong baseline. Most of the other FMTL algorithms require the presence of a parameter server, which is why we restrict the comparison to dFedU. Given that both dFedU and Sheaf-FMTL operate in decentralized settings without relying on a central server, dFedU provides a pertinent point of comparison. Finally, we are willing to compare to other FMTL algorithms in the decentralized setting if the reviewer thinks that such algorithms exist.
>
> > If considered for larger rounds, will the performance of solving the problems individually have a similar MSE in Fig 3b? If so, given that this also saves the communication cost, what is the advantage of the proposed approach over the naive approach of solving the tasks separately?
>
> **Response:**
>
> We thank the reviewer for the question. We have **updated Fig 3b** by running for more rounds. We can still see that our approach still outperforms the naive approach as highlighted in Fig 3a also.

---

> ### Author Response · Authors · 2024-11-21
> **Response to Reviewer NGQr (Part 4)**
>
> > The theoretical result on bounded gradient is not very significant under the assumptions on smoothness in the paper. The significance of the proposed approach is unclear in the absence of a regret bound.
>
> **Response:**
>
> Thank you for your  feedback and for highlighting the importance of establishing a regret bound for our proposed approach. We appreciate the opportunity to clarify the significance of our work in the context of existing FL frameworks.
>
> **Clarification of the Learning Setting**
>
> Our work primarily addresses **Federated Multi-Task Learning (FMTL)** in a **decentralized and offline** setting. In this context, each client possesses a static dataset at the outset of the machine learning (ML) optimization process. Unlike **Online Federated Learning (OFL)**, where data arrives in a sequential (streaming) manner and models are updated in real-time with each new data point, our approach assumes that all data samples are available to the clients from the beginning of the training phase. This distinction is crucial because the evaluation metrics and optimization objectives differ between offline and online settings.
>
> **Convergence vs. Regret in Offline FMTL**
>
> In traditional FL and FMTL scenarios, especially within an **offline learning paradigm**, the primary objective is to optimize a global or personalized model by collaboratively minimizing a predefined loss function across all clients. The most relevant performance metrics in this setting are related to **convergence** properties, such as:
>
> - **Convergence to Stationary Points:** Ensuring that the iterative optimization process approaches points where the gradient of the loss function is near zero.
> - **Convergence Rates:** Quantifying how quickly the algorithm approaches these stationary points, often expressed in terms of the number of iterations or communication rounds.
>
> Our **Theorem 1** establishes a **sublinear convergence rate** for our proposed algorithm, Sheaf-FMTL, demonstrating that the **average gradient norm** diminishes proportionally to $\mathcal{O}\left(\frac{1}{K}\right)$. This result is significant as it aligns our algorithm's performance with state-of-the-art decentralized FMTL algorithms, ensuring efficiency and scalability in converging to optimal models.
>
> **Regret Bounds in Online Federated Learning**
>
> Regret bounds are a cornerstone of **Online Learning (OL)** and, by extension, **Online Federated Learning (OFL)**. They measure the cumulative difference in performance between the online algorithm and the best possible fixed model in hindsight over a sequence of learning steps. Regret analysis is particularly pertinent in scenarios where models must adapt continuously to new data streams, emphasizing the algorithm's ability to perform well over time without prior knowledge of future data.
>
> **Why a Regret Bound is Not Directly Applicable to Our Work**
>
> Given that our work operates within an **offline FMTL framework**, where all data is available at the onset and models are trained in a batch-like manner, the concept of regret—fundamental to online settings—does not directly apply. Instead, our focus on **convergence guarantees** provides a robust measure of the algorithm's ability to optimize the loss function effectively over fixed datasets.
>
> However, we acknowledge that bridging offline FMTL with online paradigms presents exciting avenues for future research. Extending Sheaf-FMTL to handle streaming data and establishing regret bounds would enhance its applicability to a broader range of real-world scenarios where data is continuously generated.

---

> ### Author Response · Authors · 2024-11-25
> **Follow-Up on Rebuttal**
>
> Dear Reviewer NGQr,
>
> One more time, we would like to express our gratitude for taking the time to review our paper. Your feedback has been extremely useful in refining and improving the work.
>
> We have carefully addressed all the comments and concerns raised in your initial review and we wanted to kindly ask if you could take another look at our responses to ensure that all your concerns have been adequately addressed. If there are any additional questions or points you'd like us to clarify, kindly let us know as the rebuttal phase is ending soon.
>
> If you find that the revisions have satisfactorily addressed your concerns, we would be grateful if you could consider updating your review score accordingly.

---

> ### Author Response · Authors · 2024-12-02
> **New baselines**
>
> First, we still believe that for a fair comparison, we need to compare our algorithm to a baseline that **leverages task diversity** and operates in **a decentralized manner without relying on a parameter server**. However, since most reviewers criticized the comparison to only dFedU, we have extended our experimental evaluation by incorporating three state-of-the-art personalized federated learning (PFL) algorithms: DITTO, pFedMe, and Per-FedAvg. These algorithms are well-regarded in the literature for addressing heterogeneity in FL environments. We conducted additional experiments on two datasets—Vehicle Sensor and HAR—to evaluate the performance of our method against these baselines. The test accuracy is averaged over 5 runs reporting both mean and standard deviation.
>
> The results in the table below demonstrate that our proposed algorithm consistently achieves the highest test accuracy, comparable to or exceeding that of dFedU, while significantly reducing the number of transmitted bits. The new baselines (DITTO, pFedMe, Per-FedAvg) achieve lower communication overheads due to their star topology, leveraging a central parameter server.
>
> | Dataset        |             Algorithm             | Test Accuracy | Transmitted Bits (kB) |
> |----------------|:---------------------------------:|:-------------:|:---------------------:|
> | Vehicle Sensor |    Sheaf-FMTL ($\gamma$ = 0.1)    |  87.6 ± 0.61  |          224          |
> |                |    Sheaf-FMTL ($\gamma$ = 0.3)    |  87.6 ± 0.52  |          672          |
> |                |               dFedU               |  87.52 ± 0.53 |         2262.4        |
> |                |               DITTO               |  84.17 ± 1.66 |         161.6         |
> |                |               pFedme              |  84.82 ± 0.65 |         161.6         |
> |                |             Per-FedAvg            |  81.85 ± 1.16 |         161.6         |
> | HAR            |    Sheaf-FMTL ($\gamma$ = 0.1)    |  92.49 ± 1.21 |          5392         |
> |                |    Sheaf-FMTL ($\gamma$ = 0.3)    |  92.82 ± 0.51 |         16176         |
> |                |               dFedU               |  93.29 ± 0.69 |         53952         |
> |                |               DITTO               |  91.86 ± 1.09 |         2697.6        |
> |                |               pFedme              |  91.63 ± 1.15 |         2697.6        |
> |                |             Per-FedAvg            |  89.73 ± 0.78 |         2697.6        |

---

### Official Review · Reviewer_svdS · 2024-11-03

**Soundness:** 3
**Presentation:** 3
**Contribution:** 2
**Rating:** 5
**Confidence:** 3

**Summary:**

This work studies the federated multi-task learning (FMTL) problem. It points out that existing FMTL frameworks have limitations in handling feature and sample heterogeneous clients and introduces a sheaf-theoretic-based method. Specifically, the optimization objective leverages a sheaf Laplacian regularization that facilitates the modeling of client model interactions. Theoretically, the proposed algorithm achieves a sublinear convergence rate and experimentally, the proposed algorithm outperforms the dFedU algorithm.

**Strengths:**

The topic is of great interest to the community. The introduction of sheaf Laplacian regulariztion to the objective function in FL is novel as it helps to model the interactions between heterogenous clients. This is meaningful when clients have different data distributions and/or model architectures.

**Weaknesses:**

The definition of federated multi-task learning (FMTL) is not clear across the manuscript and it seems that there exists disparity between what the authors claim it can do and what the method actually does in experiments. In the introduction, it is mentioned that "FMTL generalizes the FL framework by allowing the learning of multiple related tasks across several clients simultaneously." In the following texts, however, it seems that the setting is narrowed down to data/model-heterogeneous federated learning. For example, the experiments considers data heterogeneity and model heterogeneity. Then it remains to be discussed how the concept of FMTL differs from personalized FL or heterogeneous FL. To handle both data and model heterogeneity in FL, there have been a series of work leveraging knowledge distillation and in which scenario, should people select the method proposed in this work?

**Questions:**

1. For high-dimensional models where $d_i$ is large, will it add communication cost and storage cost when messages are transmitted among the network? At each iteration $k$, each client needs to communicate twice to update the model and the matrices separately.

2. In the experimental section, more baseline algorithms should be compared. Currently the only benchmarking algorithm is dFedU algorithm. More personalized FL algorithms should be included here (some are included in the paper of dFedU algorithm).

3. The difference between FMTL and conventional personalized FL/heterogeneous FL should be elaborated in more details.

---

> ### Author Response · Authors · 2024-11-20
> **Response to Reviewer svdS**
>
> > **Weaknesses**
>
> **Response:**
>
> We appreciate the reviewer's feedback and understand the importance of having a precise and consistent characterization of FMTL to better highlight the contributions and applicability of our proposed Sheaf-FMTL framework. **In response, we have added more detail in the introduction and related work sections. We also added Appendix C to differentiate FMTL from PFL and HFL in terms of objective, task diversity, model architecture, and knowledge-sharing mechanism.**
>
> > To handle both data and model heterogeneity in FL, there have been a series of work leveraging knowledge distillation and in which scenario, should people select the method proposed in this work?
>
> **Response:**
>
> We thank the reviewer for this question. Sheaf-FMTL offers distinct advantages over knowledge distillation (KD) based approaches in FL, particularly in scenarios involving both data and model heterogeneity. Unlike KD methods that primarily facilitate knowledge transfer between model architectures, Sheaf-FMTL dynamically learns and models complex interactions between client models through learned restriction maps, enabling a more structured and flexible accommodation of task relationships and model differences. Additionally, Sheaf-FMTL integrates these interactions directly into a unified optimization framework via sheaf Laplacian regularization, allowing simultaneous optimization of individual models and their interdependencies. This approach not only enhances communication efficiency by transmitting lower-dimensional projections of model parameters but also preserves model flexibility, making it ideal for multi-task environments with diverse model architectures. Therefore, Sheaf-FMTL is particularly recommended for federated networks characterized by multi-task learning needs, heterogeneous model structures, and sparse and structured client interactions. In contrast, KD-based methods are preferable when addressing model heterogeneity without explicit multi-task relationships or when bi-directional knowledge transfer between models is essential.
>
> > For high-dimensional models where $d_i$ is large, will it add communication cost and storage cost when messages are transmitted among the network? At each iteration $k$, each client needs to communicate twice to update the model and the matrices separately.
>
> **Response:**
>
> We agree with the reviewer that in high-dimensional settings, Sheaf-FMTL introduces additional communication and storage overheads due to the transmission of both model projections and restriction maps. However, these costs are mitigated by several factors. Firstly, the communication payload for Sheaf-FMTL involves transmitting lower-dimensional projections ($d_{ij} \ll d_i$), which substantially reduces the overall bandwidth requirements compared to sending full model parameters. Secondly, the storage overhead associated with restriction maps is manageable by selecting a small projection dimension $d_{ij}$, thereby keeping the additional storage proportional to $d_i \times d_{ij}$. **Furthermore, as detailed in Appendix E, optimization strategies such as dimensionality reduction, sparse restriction maps, and efficient matrix operations can significantly alleviate both computational and storage burdens.** These measures ensure that Sheaf-FMTL remains scalable and efficient even when handling complex, high-dimensional models in FL environments.
>
> > In the experimental section, more baseline algorithms should be compared. Currently the only benchmarking algorithm is dFedU algorithm. More personalized FL algorithms should be included here (some are included in the paper of dFedU algorithm).
>
> **Response:**
>
> We appreciate the reviewer's observation regarding the number of methods used for comparison in our study. To the best of our knowledge, dFedU is the state-of-the-art method in FMTL in the decentralized setting, and hence it serves as a strong baseline. Most of the other FMTL algorithms require the presence of a parameter server, which is why we restrict the comparison to dFedU. Given that both dFedU and Sheaf-FMTL operate in decentralized settings without relying on a central server, dFedU provides a pertinent point of comparison. Note that the baselines mentioned in the experimental section of the dFedU paper concern the FedU version of the algorithm, which relies on a parameter server. Finally, we are willing to compare to other FMTL algorithms in the decentralized setting if the reviewer thinks that such algorithms exist.

---

> > ### Comment · Reviewer_svdS · 2024-11-28
> >
> > Thanks for the responses. I have read the responses as well as the updated manuscript. If I understand correctly, the task diversity is a unique part in FMTL, compared to most personalized FL or heterogeneous FL work. If this is the case, then the experiment design is not solid enough since the task diversity is not reflected and only data/model heterogeneity is considered.
> >
> > Moreover, regarding the comment on baseline algorithms, the point is that since the experiments focus on data and model heterogeneity, it is expected that the authors compare proposed algorithm with those designed for heterogeneous FL. There could be another set of experiments for task diversity that only include dFedU algorithm.

---

> > > ### Author Response · Authors · 2024-11-28
> > > **Response to further discussion (Part 1)**
> > >
> > > We thank the reviewer for reading our response and the revised manuscript. We acknowledge the reviewer’s observation that task diversity is a pivotal aspect of FMTL. To address the concern that our experiments primarily showcase data/model heterogeneity without explicitly reflecting task diversity, we emphasize that task diversity in our experiments is intrinsically embedded within the multi-dataset and multi-client configurations as illustrated in several SOTA FMTL works [Ref1-Ref3]. Each dataset and its corresponding client partitioning are designed to simulate distinct learning tasks or subtasks, thereby aligning with the essence of task diversity in FMTL. In fact, our experimental setup incorporates task diversity through the utilization of multiple datasets, each representing distinct learning tasks across different domains. Here is a detailed breakdown of how each dataset embodies task diversity:
> > >
> > > - **Rotated MNIST (R-MNIST):**
> > >   - **Task Description:** While all clients participate in the digit classification task, each group of clients deals with images rotated by distinct angles (0°, 90°, 180°, 270°).
> > >   - **Task Diversity Aspect:** Despite sharing the same overarching classification objective, the rotation-induced variations introduce distinct sub-tasks within the same dataset. Each rotation angle can be perceived as a separate but related task, requiring the model to adapt to different orientations of the input data.
> > >
> > > - **Heterogeneous CIFAR-10 (H-CIFAR-10):**
> > >   - **Task Description:** Each client is assigned a classification task involving a subset of 5 randomly selected classes out of the 10 available in CIFAR-10.
> > >   - **Task Diversity Aspect:** This partitioning creates multiple classification sub-tasks across clients, each focusing on a different combination of classes. These varied class assignments simulate distinct classification challenges, thereby introducing task diversity.
> > >
> > > - **Human Activity Recognition (HAR):**
> > >   - **Task Description:** Each client focuses on distinguishing between six different human activities based on sensor data.
> > >   - **Task Diversity Aspect:** Although the task is classification, the individual variability in sensor data across clients (e.g., differences in user behavior, sensor placement, environmental factors) introduces unique subtasks that require the model to generalize across diverse activity patterns.
> > >
> > > - **Vehicle Sensor:**
> > >   - **Task Description:** Clients are tasked with the binary classification of vehicles (Assault Amphibian Vehicle vs. Dragon Wagon) based on multi-sensor data.
> > >   - **Task Diversity Aspect:** Variations in sensor types (acoustic, seismic, infrared) and their respective data characteristics across different clients introduce distinct learning challenges, effectively creating diverse classification subtasks.
> > >
> > > - **Google Glass Eating and Motion (GLEAM):**
> > >   - **Task Description:** Clients engage in binary classification to determine if the wearer is eating based on high-resolution sensor data.
> > >   - **Task Diversity Aspect:** Differences in individual behaviors, sensor calibration, and contextual environments across users lead to unique classification subtasks, fostering task diversity.
> > >
> > > - **School:**
> > >   - **Task Description:** Clients predict exam results for students based on school-specific and student-specific features.
> > >   - **Task Diversity Aspect:** Variations in school environments, student demographics, and educational resources create distinct regression subtasks that the model must address, embodying task diversity.
> > > ---
> > > [Ref1] Smith, V., Chiang, C. K., Sanjabi, M., and Talwalkar, A. S.. "Federated multi-task learning", Advances in neural information processing systems, 30, 2017.
> > >
> > > [Ref2] Dinh, Canh T., Tung T. Vu, Nguyen H. Tran, Minh N. Dao, and Hongyu Zhang. "A new look and convergence rate of federated multitask learning with laplacian regularization." IEEE Transactions on Neural Networks and Learning Systems, 2022.
> > >
> > > [Ref3] Liu, K., Hu, S., Wu, S. Z., & Smith, V. (2022). On privacy and personalization in cross-silo federated learning. Advances in neural information processing systems, 35, 5925-5940.

---

> > > ### Author Response · Authors · 2024-11-28
> > > **Response to further discussion (Part 2)**
> > >
> > > We acknowledge the reviewer's suggestion to compare our algorithm with existing PFL/HFL algorithms. However, to the best of our knowledge, most established PFL/HFL baselines are designed for a star-topology network with a central server, limiting their direct applicability to fully decentralized settings with random topologies. Since this is a common concern to all reviewers, we will add a table to compare our algorithm to some PFL/HFL baselines (for star topology) for some selected datasets. We will update our response with the table once they are ready.

---

> ### Author Response · Authors · 2024-11-25
> **Follow-Up on Rebuttal**
>
> Dear Reviewer svdS,
>
> One more time, we would like to express our gratitude for taking the time to review our paper. Your feedback has been extremely useful in refining and improving the work.
>
> We have carefully addressed all the comments and concerns raised in your initial review and we wanted to kindly ask if you could take another look at our responses to ensure that all your concerns have been adequately addressed. If there are any additional questions or points you'd like us to clarify, kindly let us know as the rebuttal phase is ending soon.
>
> If you find that the revisions have satisfactorily addressed your concerns, we would be grateful if you could consider updating your review score accordingly.

---

> ### Author Response · Authors · 2024-12-02
> **Response to further discussion (Part 3)**
>
> First, we still believe that for a fair comparison, we need to compare our algorithm to a baseline that **leverages task diversity** and operates in **a decentralized manner without relying on a parameter server**. However, since most reviewers criticized the comparison to only dFedU, we have extended our experimental evaluation by incorporating three state-of-the-art personalized federated learning (PFL) algorithms: DITTO, pFedMe, and Per-FedAvg. These algorithms are well-regarded in the literature for addressing heterogeneity in FL environments. We conducted additional experiments on two datasets—Vehicle Sensor and HAR—to evaluate the performance of our method against these baselines. The test accuracy is averaged over 5 runs reporting both mean and standard deviation.
>
> The results in the table below demonstrate that our proposed algorithm consistently achieves the highest test accuracy, comparable to or exceeding that of dFedU, while significantly reducing the number of transmitted bits. The new baselines (DITTO, pFedMe, Per-FedAvg) achieve lower communication overheads due to their star topology, leveraging a central parameter server.
>
> | Dataset        |             Algorithm             | Test Accuracy | Transmitted Bits (kB) |
> |----------------|:---------------------------------:|:-------------:|:---------------------:|
> | Vehicle Sensor |    Sheaf-FMTL ($\gamma$ = 0.1)    |  87.6 ± 0.61  |          224          |
> |                |    Sheaf-FMTL ($\gamma$ = 0.3)    |  87.6 ± 0.52  |          672          |
> |                |               dFedU               |  87.52 ± 0.53 |         2262.4        |
> |                |               DITTO               |  84.17 ± 1.66 |         161.6         |
> |                |               pFedme              |  84.82 ± 0.65 |         161.6         |
> |                |             Per-FedAvg            |  81.85 ± 1.16 |         161.6         |
> | HAR            |    Sheaf-FMTL ($\gamma$ = 0.1)    |  92.49 ± 1.21 |          5392         |
> |                |    Sheaf-FMTL ($\gamma$ = 0.3)    |  92.82 ± 0.51 |         16176         |
> |                |               dFedU               |  93.29 ± 0.69 |         53952         |
> |                |               DITTO               |  91.86 ± 1.09 |         2697.6        |
> |                |               pFedme              |  91.63 ± 1.15 |         2697.6        |
> |                |             Per-FedAvg            |  89.73 ± 0.78 |         2697.6        |

---

### Official Review · Reviewer_R4pY · 2024-11-03

**Soundness:** 2
**Presentation:** 3
**Contribution:** 3
**Rating:** 6
**Confidence:** 3

**Summary:**

This paper introduces a novel framework for Federated Multi-Task Learning (FMTL) that addresses the challenges of feature and sample heterogeneity across clients in a decentralized setting. A Sheaf-based approach Sheaf-FMTL is used to model complex interactions between client models, promoting local communication while maintaining performance. Experiments demonstrate that Sheaf-FMTL can support communication with models of different sizes and reduce the communication burden.

Overall, this article gives an effective way of organizing federated clients that achieves good inter-client information communication and could be a direction for federated learning training and development. However, there may be some parts that need to be further described, clarified and supplemented.

**Strengths:**

Overall, this article gives an effective way to organize federated clients that achieves good inter-client information communication and could be a direction for federated learning training and development.

**Weaknesses:**

For the theoretical part, the objectives formulation of FMTL in existing scenarios should be clarified before elaborating the methodology Sheaf-FMTL. To support the claimed contribution that the Sheaf-FMTL design unifies PERSONALIZED FL, CONVENTIONAL FMTL, HYBRID FL, and CONVENTIONAL FL, more illustrations combined with formulas in addition to Remark 3.3 are supposed to provide.

On the experimental side, I currently have the following questions and concerns,
(1) the additional time-space costs associated with the training and storage of restriction maps need to be clarified.
(2) Can Sheaf-FMTL adapt to more general FL training scenarios? The authors are suggested to explain the scalability of the proposed method, especially, with more complex data, and larger parameter space (e.g. more complex neural networks). Also, additional computational costs, time and storage space consumption are the concerns under the complex scenarios.
(3) Since this work aims to address heterogeneity and unify multiple FL scenarios, data partitioning should be made more explicit, either in terms of the amount of data, domains (maybe), and classes (label distribution) between clients, or generalized by some sort of Non-IID metrics or pre-existing partitioning instance.

**Questions:**

1. Please clarify the additional time-space costs associated with the training and storage of restriction maps.
2. Can Sheaf-FMTL adapt to more general FL training scenarios? Please explain the scalability of the proposed method, especially, with more complex data, and larger parameter space (e.g. more complex neural networks).
3. Please give more explicit data partitioning.

---

> ### Author Response · Authors · 2024-11-20
> **Response to Reviewer R4pY**
>
> > To support the claimed contribution that the Sheaf-FMTL design unifies PERSONALIZED FL, CONVENTIONAL FMTL, HYBRID FL, and CONVENTIONAL FL, more illustrations combined with formulas in addition to Remark 3.3 are supposed to provide.
>
> **Response:**
>
> We thank the reviewer for this comment. To address the concern and strengthen our claim that the Sheaf-FMTL framework unifies Personalized FL, Conventional FMTL, Hybrid FL, and Conventional FL, **we have expanded the mathematical explanation in Appendix C**. We also added **Table 2** which summarizes how different FL paradigms are special cases of the Sheaf-FMTL framework based on the choice of restriction maps.
>
> > The additional time-space costs associated with the training and storage of restriction maps need to be clarified.
>
> **Response:**
>
> We appreciate the reviewer’s comment regarding discussing the additional time and space costs introduced by the restriction maps in our Sheaf-FMTL framework. **To address this, we have incorporated a detailed comparative analysis in our manuscript, as can be seen in Table 1**. This table summarizes the storage requirements, computational overheads, and communication savings per client and per iteration for both Sheaf-FMTL and the dFedU baseline.
>
> > Can Sheaf-FMTL adapt to more general FL training scenarios? The authors are suggested to explain the scalability of the proposed method, especially, with more complex data, and larger parameter space (e.g. more complex neural networks). Also, additional computational costs, time and storage space consumption are the concerns under the complex scenarios.
>
> **Response:**
>
> Thank you for the feedback regarding the scalability of our proposed Sheaf-FMTL method in more general FL scenarios involving complex data and larger parameter spaces. To address your concerns, we have conducted a comprehensive analysis of the storage, computational overheads, and communication costs associated with Sheaf-FMTL in comparison to the baseline dFedU, as detailed in **Table 1**. Our findings indicate that while Sheaf-FMTL does incur additional storage and computational costs due to the maintenance and training of restriction maps, these are significantly offset by the substantial communication savings achieved, especially when the projection space dimension $d_{ij}$ is chosen to be a small fraction of the model dimension $d_i$. Furthermore, in the broader context of distributed learning, there is often a tradeoff between communication, computation, and storage. Our method prioritizes communication efficiency, which is often the bottleneck in FL, at the cost of additional storage and compute. This trade-off makes Sheaf-FMTL highly viable in resource-rich FL environments, such as cross-silo settings. **A detailed discussion on the scalability of Sheaf-FMTL, including its application to more complex data and larger parameter spaces as well as some strategies to mitigate the associated computational and storage overheads, is provided in Appendix E.**
>
> > Since this work aims to address heterogeneity and unify multiple FL scenarios, data partitioning should be made more explicit, either in terms of the amount of data, domains (maybe), and classes (label distribution) between clients, or generalized by some sort of Non-IID metrics or pre-existing partitioning instance.
>
> **Response:**
>
> Thank you for your feedback and for emphasizing the importance of explicitly detailing the data partitioning strategies used in our experiments. We understand that clear and comprehensive descriptions of data distribution are crucial for evaluating the effectiveness and generalizability of our Sheaf-FMTL framework, especially in heterogeneous FL scenarios. To this end, we added **two tables**: (i) **Table 4** details the data partitioning strategies across datasets by specifying the minimum and maximum number of training samples per client, the domain and label distributions, and (ii) **Table 5** summarizes the degree of non-IIDness across datasets and the quantitative metrics assessing the degree of non-IIDness.

---

> ### Author Response · Authors · 2024-11-25
> **Follow-Up on Rebuttal**
>
> Dear Reviewer R4pY,
>
> One more time, we would like to express our gratitude for taking the time to review our paper. Your feedback has been extremely useful in refining and improving the work.
>
> We have carefully addressed all the comments and concerns raised in your initial review and we wanted to kindly ask if you could take another look at our responses to ensure that all your concerns have been adequately addressed. If there are any additional questions or points you'd like us to clarify, kindly let us know as the rebuttal phase is ending soon.
>
> If you find that the revisions have satisfactorily addressed your concerns, we would be grateful if you could consider updating your review score accordingly.

---

### Author Response · Authors · 2024-11-20
**Global Response**

We thank the reviewers for taking the time to read our paper and for their detailed feedback and suggestions for improvement. We revised the manuscript as per your suggestions, with all changes highlighted in blue in the updated file. For convenience, we list these changes below:
- We enhanced the mathematical explanation in **Appendix D** to demonstrate how the Sheaf-FMTL framework unifies Personalized FL, Conventional FMTL, Hybrid FL, and Conventional FL.
- We included **Table 2**, which summarizes how different FL paradigms are special cases of the Sheaf-FMTL framework based on the choice of restriction maps.
- We incorporated a detailed comparative analysis in the manuscript, presented in **Table 1**, summarizing the storage requirements, computational overheads, and communication savings per client and per iteration for both Sheaf-FMTL and the dFedU baseline.
- We provided a comprehensive discussion on the scalability of Sheaf-FMTL, including its application to more complex data and larger parameter spaces, as well as strategies to mitigate associated computational and storage overheads, in **Appendix E**.
- We detailed data partitioning strategies in **Tables 4 and 5**.
- We included details in **Appendix C** to differentiate FMTL from PFL and HFL in terms of objective, task diversity, model architecture, and knowledge-sharing mechanism.
- We added **Appendix F** to mention two real-world examples where task similarities can be modelled using vector spaces.
- We updated  **Figure 3b** by running for more rounds.
- We added  **a paragraph** after Lemma 3.2 in the revised manuscript to specify the application of sheaf theory in FMTL.
- We added **a paragraph** before Eq. (8) to motivate and provide intuition on its derivation.
- Based on the reviewers' suggestion, we have added three state-of-the-art personalized federated learning (PFL) algorithms: DITTO, pFedMe, and Per-FedAvg. This addition is shown only in the response below and is not incorporated in the revised version since it was added after Nov. 27th.

In addition to submitting a revised version, we reply to each reviewer individually to address their questions and concerns. We hope that these responses together with the revised manuscript clear up any confusion and resolve all issues that the reviewers had, and that you will consider increasing your rating for our paper.

---

### Meta-Review · Area_Chair_mrMB · 2024-12-20

**Metareview:**

The paper proposes a sheaf-theoretic framework for federated multi-task learning to address client heterogeneity in decentralized settings. Using sheaf Laplacian regularization, the method models complex client-task interactions and achieves efficient decentralized learning with privacy preservation. Contributions include theoretical convergence guarantees, compatibility with diverse methods, and improved communication efficiency validated through extensive experiments.

The paper failed to effectively communicate its underlying concepts to the reader. Several reviewers mentioned this issue and the need for a clearer story to justify the proposed methodology. It was also noted that some additional experiments are needed.

**Additional Comments On Reviewer Discussion:**

R4pY noted the need for clarification of the study's scenario. svdS and RmE8 similarly questioned the scenario and the importance of the method. svdS and RmE8 and NGQr additionally pointed out the inadequacy of the comparison experiments. The authors offered counterarguments, but they failed to change the opinion of the reviewers.

---

### Decision · Program_Chairs · 2025-01-22

Reject